# A Unified Analysis of Federated Learning with Arbitrary Client Participation

**Shiqiang Wang**
IBM T. J. Watson Research Center
Yorktown Heights, NY 10598
wangshiq@us.ibm.com

**Mingyue Ji**
Department of ECE, University of Utah
Salt Lake City, UT 84112
mingyue.ji@utah.edu

## Abstract

Federated learning (FL) faces challenges of intermittent client availability and computation/communication efficiency. As a result, only a small subset of clients can participate in FL at a given time. It is important to understand how partial client participation affects convergence, but most existing works have either considered idealized participation patterns or obtained results with non-zero optimality error for generic patterns. In this paper, we provide a unified convergence analysis for FL with arbitrary client participation. We first introduce a generalized version of federated averaging (FedAvg) that amplifies parameter updates at an interval of multiple FL rounds. Then, we present a novel analysis that captures the effect of client participation in a single term. By analyzing this term, we obtain convergence upper bounds for a wide range of participation patterns, including both non-stochastic and stochastic cases, which match either the lower bound of stochastic gradient descent (SGD) or the state-of-the-art results in specific settings. We also discuss various insights, recommendations, and experimental results.

## 1 Introduction

We consider a federated learning (FL) problem with $N$ clients [17, 20, 38]. Our goal is to minimize:

$$f(\mathbf{x}) := \frac{1}{N} \sum_{n=1}^{N} F_n(\mathbf{x}), \tag{1}$$

where $\mathbf{x} \in \mathbb{R}^m$ is an $m$-dimensional model parameter. The local objective $F_n(\mathbf{x})$ of each client $n$ is usually defined as $F_n(\mathbf{x}) := \mathbb{E}_{\xi_n \sim \mathcal{D}_n} [\ell_n(\mathbf{x}, \xi_n)]$, where $\mathcal{D}_n$ is the distribution of client $n$'s local data, which cannot be observed globally because the data remains private, and $\ell_n(\mathbf{x}, \xi)$ is the per-sample loss function for parameter $\mathbf{x}$ and input data $\xi$. The objective (1) can be extended to a weighted average, but we do not write out the weights and consider them as part of $\ell_n(\mathbf{x}, \xi)$ and $F_n(\mathbf{x})$. A canonical way of solving (1) is to use federated averaging (FedAvg) [24], which is a form of stochastic gradient descent (SGD) that operates in multiple rounds, where each *round* includes multiple local update steps followed by communication between clients and a server to synchronize the updates.

**Partial Participation.** A major challenge in FL is that clients only intermittently participate in the collaborative training process [3]. The reason is twofold. First, FL applications usually have a large pool of clients. It is impractical to have all clients participating in all FL rounds, because it would require computation and communication that can consume a large amount of energy and also lead to network congestion. Second, clients may become unavailable from time to time. For example, if the client devices are smartphones, they may be willing to participate in FL only when they are charging (usually at night), to avoid draining the battery in outdoor environments where no charger is available.

**Open Problem.** The convergence of FL algorithms with realistic client participation patterns has not been fully understood. Most existing works on partial participation have only considered idealized scenarios, such as those where clients are randomly selected according to a given probability distribution that is independent across rounds [4, 8, 9, 18, 21, 22, 36]. They cannot incorporate cases

36th Conference on Neural Information Processing Systems (NeurIPS 2022).

Table 1: Summary and comparison of results (only showing baselines without constant error term)

| Method | Participation | Convergence error upper bound | Remark |
|---|---|---|---|
| Yan et al. [35] | Unavailable $\leq E$ rounds | $\mathcal{O}\left(\frac{(1+\sigma^2)N^{1/4}\sqrt{E}}{\sqrt{ST}}\right)$ | *Add'l assumption:* bounded gradient norm |
| Gu et al. [12] | Unavailable $\leq E$ rounds | $\mathcal{O}\left(\frac{(1+\sigma^2)\sqrt{E}}{\sqrt{NIT}}\right)$ | *Add'l assumptions:* Hessian Lipschitz, a.s. bounded noise |
| **Ours** (this paper) | Regularized | $\mathcal{O}\left(\frac{\sigma}{\sqrt{SIT}}\right)$ | Cor. 4.3, matching centralized SGD lower bound |
| | Ergodic | Approaches zero as $T \to \infty$ | Prop. 4.4 |
| | Mixing | $\mathcal{O}\left(\frac{1+\sigma^2}{\sqrt{SIT}}\right)$ i.E.; $\mathcal{O}\left(\frac{1+\sigma^2}{c\sqrt{SIT}}\right)$ w.p. $1-c$ | Prop. 4.6 & Cor. 4.8, matching idealized bound |
| | Independent | $\mathcal{O}\left(\frac{1+\sigma^2}{\sqrt{SIT}}\right)$ i.E.; $\mathcal{O}\left(\frac{(1+\sigma^2)\ln(2/c)}{\sqrt{SIT}}\right)$ w.p. $1-c$ | Prop. 4.7 & Cor. 4.8, matching idealized bound |

Note: *We let $\rho = \frac{1}{\sqrt{S}}$ in our results for a direct comparison with other works*, see Section 5 for more details. We consider sufficiently large $T$ in all cases and only show the dominant term in this table, with other variables ignored in $\mathcal{O}(\cdot)$. Definitions: i.E. = in expectation, w.p. = with probability, a.s. = almost surely, $S$: number of participating clients in each round ($S \leq N$), other definitions are given throughout the paper.

where clients are unavailable for a period of time. Some recent results considering general scenarios either include a constant error term that does not vanish at convergence [5, 37] or are otherwise less competitive than state-of-the-art results obtained in idealized settings [12, 35]. Therefore, we ask:
*1. What classes of participation patterns have guaranteed convergence to zero error?*
*2. For these classes, can the convergence rate match with that obtained in idealized scenarios?*

To answer these questions, we need to overcome two challenges. First, we need to devise a *unified analytical methodology* for obtaining the convergence rates under various participation patterns. Second, we need a *general and realistic algorithm* for cross-device FL, with appropriate control options that can be configured to make the convergence rate competitive.

**Our Contributions.** In this paper, we overcome these challenges and give answers to the two questions above. We first introduce a generalized version of FedAvg, which *amplifies* the parameter updates after every $P$ rounds, for some $P \geq 1$. When $P = 1$, our setting is the same as FedAvg with two-sided learning rates [18, 36], but we focus on an extended setup where $P$ can be greater than one. For this generalized FedAvg algorithm, we perform a new convergence analysis that *unifies* the effect of partial participation into a single term ($\tilde{\delta}^2(P)$ in Section 3). This allows us to *decouple* our main convergence analysis from the analysis on partial participation, which largely simplifies the analytical procedure. Our novel methodology gives the results shown in Table 1, for a broad range of participation patterns. These patterns include *non-stochastic and regularized* participation, where all clients participate equally but not simultaneously within $P$ rounds, and *stochastic* participation settings including *ergodic*, *stationary and strongly mixing* (e.g., Markov process), and *independent*. For the stochastic participation, we provide both expected and high probability convergence bounds, where the expected rates match the state-of-the-art FedAvg convergence rates that were derived for the idealized setting of random participation following an independent distribution. Furthermore, we provide new insights related to non-IID data, amplification of updates, "linear speedup" [39], etc.

To summarize, our main contributions are as follows:

- We introduce a generalized FedAvg algorithm which amplifies the updates aggregated over multiple rounds, where the amplification interval $P$ can be tuned for the best convergence.
- We present novel analysis and unified methodology for obtaining competitive convergence upper bounds with arbitrary client participation patterns.
- We discuss important insights from both the theoretical and experimental results.

**Related Work.** FedAvg [24] is characterized by partial client participation and multiple local updates in each round. In the case of full participation, FedAvg is also known as local SGD [10, 14, 23, 28–30] and parallel restarted SGD [40]. Partial participation was considered in [4, 8, 9, 18, 21, 22, 36], with the assumption that clients are selected to participate probabilistically according to a given distribution that is independent across rounds. Some recent works have started to incorporate unavailable clients. For example, the work in [27] allows inactive clients, but the theoretical result does not guarantee

convergence if a client participates with zero probability in a certain round. Similarly, the result in [5, 37] includes a constant error term in the case of general participation in cross-device FL. Other works have obtained convergence rates related to the maximum number of inactive rounds, based on additional assumptions such as Hessian Lipschitz [12] and bounded gradient norm [35]. Their theory also requires all clients to participate once at the beginning (see [12, Remark 5.2]). In contrast, our result in this paper is based on a minimal set of assumptions. We consider a simple algorithm without requiring initial full participation, and we provide a better rate of convergence to zero error (see Table 1). Moreover, existing methods usually have separate analyses and results for each specific class of participation pattern, whereas we present a unified framework that largely simplifies the analytical procedure. Some more discussions on related works are in Appendix A. Note that *all the appendices of this paper are in the supplementary material.*

## 2 Generalized FedAvg with Amplified Updates

We consider a generalized version of FedAvg as shown in Algorithm 1. In this algorithm, we define $q_t^n$ to be the *participation weight* of client $n$ in round $t$. If $q_t^n = 0$, the client does not participate in this round. Each client $n$ participates whenever $q_t^n > 0$, where the weight $q_t^n$ is applied in the global update step in Line 9. For mathematical convenience in our analysis, we assume in Algorithm 1 that all clients compute their local updates in Lines 4–8, regardless of whether they participate in the current round or not. We emphasize that this is *logically equivalent* to the practical setting where non-participating clients do not compute, because their computations have no effect in subsequent rounds when $q_t^n = 0$ (see Line 9). In every round $t$, each client $n$ computes $I$ steps of local updates according to Line 7, where $\gamma > 0$ is the local learning rate and $\mathbf{g}_n(\mathbf{y}_{t,i}^n)$ is the stochastic gradient of $F_n(\mathbf{y}_{t,i}^n)$ such that $\mathbb{E}\left[\mathbf{g}_n(\mathbf{y}_{t,i}^n) \middle| \mathbf{y}_{t,i}^n\right] = \nabla F_n(\mathbf{y}_{t,i}^n)$.

Starting at round $t_0$, the global updates are accumulated in $\mathbf{u}$ for $P$ rounds (Line 10). At the end of every interval of $P$ rounds, an amplification is applied to the accumulated updates (Line 12). Since $\mathbf{x}_t$ is first updated without amplification in rounds $t_0, \ldots, t_0 + P - 1$, Line 12 adds $\mathbf{u}$ after multiplying $\eta - 1$, which is equivalent to setting $\mathbf{x}_{t+1}$ to $\mathbf{x}_{t_0} + \eta\mathbf{u}$. When $\eta = 1$, there is no amplification and Line 12 has no effect in this case. In general, we allow any $\eta > 0$ including values of $\eta$ that are less than one. When $\eta < 1$, we reduce (instead of amplify) the updates. Intuitively, amplification with $\eta > 1$ is usually preferred over reduction, because we can compute the gradients on similar model parameters in each round where only a small number of clients participates. After $P$ rounds, the accumulated updates provide a better estimation for the overall client population, and amplifying the updates allows the model parameter to progress towards the descent direction for the majority of clients (see Appendix B.1 for an example). Also note that although $P$ is a parameter in our algorithm, the dominant terms of our convergence upper bounds shown in Table 1 *do not* depend on $P$.

---

**Algorithm 1:** Generalized FedAvg with amplified updates and arbitrary participation

---
1   **Input:** $\gamma, \eta, \mathbf{x}_0, I, P, T$;   **Output:** $\{\mathbf{x}_t : \forall t\}$
2   Initialize $t_0 \leftarrow 0$, $\mathbf{u} \leftarrow \mathbf{0}$;
3   **for** $t = 0, \ldots, T - 1$ **do**
4     **for** $n = 1, \ldots, N$ *in parallel* **do**
5       $\mathbf{y}_{t,0}^n \leftarrow \mathbf{x}_t$;
6       **for** $i = 0, \ldots, I - 1$ **do**
7         $\mathbf{y}_{t,i+1}^n \leftarrow \mathbf{y}_{t,i}^n - \gamma\mathbf{g}_n(\mathbf{y}_{t,i}^n)$
8       $\Delta_t^n \leftarrow \mathbf{y}_{t,I}^n - \mathbf{x}_t$;

9   $\mathbf{x}_{t+1} \leftarrow \mathbf{x}_t + \sum_{n=1}^N q_t^n \Delta_t^n$;   //update
10   $\mathbf{u} \leftarrow \mathbf{u} + \sum_{n=1}^N q_t^n \Delta_t^n$;   //accumulate
11   **if** $t + 1 - t_0 = P$ **then**
12     $\mathbf{x}_{t+1} \leftarrow \mathbf{x}_{t+1} + (\eta - 1)\mathbf{u}$;   //amplify
13     $t_0 \leftarrow t + 1$;
14     $\mathbf{u} \leftarrow \mathbf{0}$;

---

## 3 Convergence Analysis and Main Result

We analyze Algorithm 1 as follows. For ease of analysis, we assume that $\sum_{n=1}^N q_t^n = 1$ for $q_t^n \geq 0$ for all $t$.[1] We also define $\rho$ such that $\sum_{n=1}^N (q_t^n)^2 \leq \rho^2$ for all $t$. As $\sum_{n=1}^N q_t^n = 1$, there always

---

[1]When $\sum_{n=1}^N q_t^n \neq 1$, the algorithm is equivalent to the case of $\{q_t^n\}$ normalized to one and a global learning rate applied to the updates in each round. Since we already amplify the updates every $P$ rounds, we do not apply a separate global learning rate to each round.

exists $\rho$ such that $\rho^2 \leq 1$. We define $\mathcal{Q} := \{q_t^n : \forall n, t\}$. Mathematically, we use the conditional expectation $\mathbb{E}[\cdot | \mathcal{Q}]$ to denote the case where $\mathcal{Q}$ is given and the expectation is only over the stochastic gradient noise. In the case where $\mathcal{Q}$ is stochastic, the full expectation $\mathbb{E}[\cdot]$ is taken over both the noise and $\mathcal{Q}$. We make the following assumptions that are commonly used in the literature.

**Assumption 1** (Lipschitz gradient).
$$\|\nabla F_n(\mathbf{x}) - \nabla F_n(\mathbf{y})\| \leq L \|\mathbf{x} - \mathbf{y}\|, \forall \mathbf{x}, \mathbf{y}, n. \tag{2}$$

**Assumption 2** (Unbiased stochastic gradient with bounded variance).
$$\mathbb{E}[\mathbf{g}_n(\mathbf{x}) | \mathbf{x}] = \nabla F_n(\mathbf{x}) \text{ and } \mathbb{E}\left[\|\mathbf{g}_n(\mathbf{x}) - \nabla F_n(\mathbf{x})\|^2 \Big| \mathbf{x}\right] \leq \sigma^2, \forall \mathbf{x}, n. \tag{3}$$

**Assumption 3** (Bounded gradient divergence).
$$\|\nabla F_n(\mathbf{x}) - \nabla f(\mathbf{x})\|^2 \leq d^2, \forall \mathbf{x}, n. \tag{4}$$

The gradient divergence in Assumption 3 is related to the degree of non-IID data distribution across clients. To better interpret how different divergence components affect the convergence, we introduce an alternative set of divergence bounds in Assumption 3' as follows. We will show later (in Section 4) that if Assumption 3 holds, then Assumption 3' also holds with properly chosen $\tilde{\beta}^2$, $\tilde{\nu}^2$, and $\tilde{\delta}^2(P)$.

**Assumption 3'** (Alternative gradient divergence bound).

$$\left\|\sum_{n=1}^{N} q_t^n [\nabla F_n(\mathbf{x}) - \nabla f(\mathbf{x})]\right\|^2 \leq \tilde{\beta}^2, \forall \mathbf{x}, t, \tag{5}$$

$$\sum_{n=1}^{N} q_t^n \left\|\nabla F_n(\mathbf{x}) - \sum_{n'=1}^{N} q_t^{n'} \nabla F_{n'}(\mathbf{x})\right\|^2 \leq \tilde{\nu}^2, \forall \mathbf{x}, t, \tag{6}$$

$$\left\|\frac{1}{P} \sum_{t=t_0}^{t_0+P-1} \sum_{n=1}^{N} q_t^n (\nabla F_n(\mathbf{x}) - \nabla f(\mathbf{x}))\right\|^2 \leq \tilde{\delta}^2(P), \forall \mathbf{x}, t_0. \tag{7}$$

In the above, $\tilde{\beta}^2$ and $\tilde{\nu}^2$ capture the gradient divergence in an arbitrary round $t$, and $\tilde{\delta}^2(P)$ captures the divergence of time-averaged gradients over $P$ rounds, which is a function of $P$. In particular, $\tilde{\beta}^2$ is an upper bound of the divergence between the original objective's gradient $\nabla f(\mathbf{x})$ and the averaged gradient among participating clients weighted by $\{q_n^t\}$. When all the clients participate, we have $q_t^n = \frac{1}{N}$ and (5) holds with $\tilde{\beta}^2 = 0$. The quantity $\tilde{\nu}^2$ is an upper bound of the divergence between each client's gradient and the averaged gradient of participating clients, where the average is also weighted by $\{q_n^t\}$. We will show in Section 4.1 that the overall divergence $d^2$ in Assumption 3 can be expressed as a sum of $\tilde{\beta}^2$ and $\tilde{\nu}^2$, as intuition suggests. The quantity $\tilde{\delta}^2(P)$ in (7) extends $\tilde{\beta}^2$ in (5) by computing the average over $P$ rounds (inside the norm) instead of a single round. When the weights $\{q_n^t\}$ are properly chosen, as $P$ gets large, the average $q_n^t$ over $P$ rounds gets close to each other for different $n$, and $\tilde{\delta}^2(P)$ becomes small. We will formally analyze this behavior in Section 4.2.

In the following, let $\mathcal{F} := f(\mathbf{x}_0) - f^*$, where $f^*$ is the true minimum of (1), i.e., $f^* := \min_{\mathbf{x}} f(\mathbf{x})$.

**Theorem 3.1.** When Assumptions 1, 2, and 3' hold, $\gamma \leq \frac{1}{12LIP}$, $\gamma\eta \leq \frac{1}{LIP}$, and $P \leq \frac{T}{2}$, we have

$$\min_t \mathbb{E}\left[\|\nabla f(\mathbf{x}_t)\|^2 \Big| \mathcal{Q}\right]$$
$$\leq \mathcal{O}\left(\frac{\mathcal{F}}{\gamma\eta IT} + \gamma^2 L^2 I^2 \tilde{\nu}^2 + \gamma^2 L^2 I^2 P^2 \tilde{\beta}^2 + \tilde{\delta}^2(P) + \left(\gamma^2 L^2 I + \gamma^2 L^2 IP\rho^2 + \gamma\eta L\rho^2\right)\sigma^2\right). \tag{8}$$

*Proof Sketch.* We first use smoothness to relate $f(\mathbf{x}_{t_0+P})$ and $f(\mathbf{x}_{t_0})$, which includes an inner-product term that can be expanded into several terms. One of these terms includes $\nabla F_n(\mathbf{y}_{t,i}^n) - \nabla f(\mathbf{x}_{t_0})$. We upper bound this term with three other terms including $\left\|\mathbf{y}_{t,i}^n - \sum_{n'=1}^{N} q_t^{n'} \mathbf{y}_{t,i}^{n'}\right\|^2$, $\left\|\sum_{n'=1}^{N} q_t^{n'} \mathbf{y}_{t,i}^{n'} - \mathbf{x}_{t_0}\right\|^2$, and $\tilde{\delta}^2(P)$. The upper bounds of the first two terms are found by solving recurrence relations, where the recurrence for the second term includes the first term. This nested recurrence is a uniqueness in this proof. The full proof is given in Appendix C. $\square$

For two specific (but different) learning rate configurations, we can obtain the following corollaries.

**Corollary 3.2.** *Choosing* $\gamma = \frac{1}{12LIP\sqrt{T}}$ *and* $\eta = \min\left\{\frac{12P\sqrt{LI\mathcal{F}}}{\sigma\rho}; 12\sqrt{T}\right\}$, *for* $P \leq \frac{T}{2}$, *we have*

$$\min_t \mathbb{E}\left[\|\nabla f(\mathbf{x}_t)\|^2 \Big| \mathcal{Q}\right] \leq \mathcal{O}\left(\frac{\sigma\rho\sqrt{L\mathcal{F}}}{\sqrt{IT}} + \frac{LP\mathcal{F}}{T} + \frac{\tilde{\nu}^2}{P^2T} + \frac{\tilde{\beta}^2}{T} + \tilde{\delta}^2(P) + \frac{\sigma^2}{IPT}\right). \quad (9)$$

**Corollary 3.3.** *If* $\frac{\sqrt{\mathcal{F}}}{\rho\sqrt{LIT}} \leq \frac{1}{LIP}$, *choosing* $\gamma = \frac{1}{12LIP\sqrt{T}}$ *and* $\eta = \frac{12P\sqrt{LI\mathcal{F}}}{\rho}$, *for* $P \leq \frac{T}{2}$, *we have*

$$\min_t \mathbb{E}\left[\|\nabla f(\mathbf{x}_t)\|^2 \Big| \mathcal{Q}\right] \leq \mathcal{O}\left(\frac{(1+\sigma^2)\rho\sqrt{L\mathcal{F}}}{\sqrt{IT}} + \frac{\tilde{\nu}^2}{P^2T} + \frac{\tilde{\beta}^2}{T} + \tilde{\delta}^2(P) + \frac{\sigma^2}{IPT}\right). \quad (10)$$

These two corollaries are key building blocks of our *unified convergence analysis framework*. The choice between them depends on whether and how fast $P$ grows with $T$. We will also see in the next section that $\tilde{\delta}^2(P)$ is key in capturing the effect of different participation patterns. After deriving $\tilde{\delta}^2(P)$ such that (7) holds, plugging it back to one of the above corollaries gives the results in Table 1.

## 4 Interpreting and Applying the Unified Framework

### 4.1 Discussion on $\tilde{\beta}^2$ and $\tilde{\nu}^2$: Decomposition of Divergence

In Corollaries 3.2 and 3.3, we have an interesting observation that the term involving $\tilde{\nu}$ decreases with $P$, whereas the term involving $\tilde{\beta}$ does not decrease with $P$. This suggests that when $P$ is large, the divergence term $\tilde{\beta}$ has the main effect on convergence. We recall that $\tilde{\nu}$ captures the divergence between the gradient of each client and the average gradient of clients that participate in a round $t$. Since the learning rate $\gamma$ is inversely proportional to both $I$ and $P$, a large $P$ gives a smaller learning rate which makes the divergence (related to $\tilde{\nu}$) after $I$ iterations in each round less significant. For the term with $\tilde{\beta}$, increasing $P$ does not have the same effect, because parameter averaging is only conducted across the participating clients instead of all the clients, and $\tilde{\beta}$ captures the divergence with respect to all the clients. The effect of all clients' participation pattern is captured by the term $\tilde{\delta}^2(P)$.

From Assumption 3, by adding and subtracting $\sum_{n'=1}^{N} q_t^{n'} \nabla F_{n'}(\mathbf{x})$ inside the norm and expanding the square, we can obtain

$$\sum_{n=1}^{N} q_t^n \|\nabla F_n(\mathbf{x}) - \nabla f(\mathbf{x})\|^2$$
$$= \sum_{n=1}^{N} q_t^n \left\|\nabla F_n(\mathbf{x}) - \sum_{n'=1}^{N} q_t^{n'} \nabla F_{n'}(\mathbf{x})\right\|^2 + \left\|\sum_{n'=1}^{N} q_t^{n'} \nabla F_{n'}(\mathbf{x}) - \nabla f(\mathbf{x})\right\|^2 \leq d^2, \quad (11)$$

because $\sum_{n=1}^{N} q_t^n = 1$ and the inner-product term is zero. The two terms in the right-hand side (RHS) of (11) are equal to the left-hand side (LHS) of (5) and (6), respectively. We thus have the following result showing that $\tilde{\beta}^2$ and $\tilde{\nu}^2$ can be seen as a decomposition of the original divergence $d^2$.

**Proposition 4.1.** *There exist $\tilde{\beta}^2$ and $\tilde{\nu}^2$, such that $\tilde{\beta}^2 + \tilde{\nu}^2 \leq d^2$ while satisfying (5) and (6).*

### 4.2 Discussion on $\tilde{\delta}^2(P)$: Effect of Partial Participation

The results in Theorem 3.1 and Corollaries 3.2 and 3.3 include a term of $\tilde{\delta}^2(P)$. To guarantee convergence to zero error, $\tilde{\delta}^2(P)$ needs to be either equal to zero or decrease in $T$.

***Condition for*** $\tilde{\delta}^2(P) = 0$. From (1), we know that $0 = \frac{1}{N}\sum_{n=1}^{N} F_n(\mathbf{x}) - f(\mathbf{x}) = \frac{1}{N}\sum_{n=1}^{N}(F_n(\mathbf{x}) - f(\mathbf{x}))$. Hence,

$$\mu_{t_0} \sum_{n=1}^{N} (F_n(\mathbf{x}) - f(\mathbf{x})) = 0 \quad (12)$$

for an arbitrary constant $\mu_{t_0}$, which can possibly depend on $t_0$. Then, we obtain

$$\left\|\frac{1}{P}\sum_{t=t_0}^{t_0+P-1}\sum_{n=1}^{N} q_t^n (\nabla F_n(\mathbf{x}) - \nabla f(\mathbf{x}))\right\|^2 = \left\|\sum_{n=1}^{N}\frac{1}{P}\sum_{t=t_0}^{t_0+P-1} q_t^n (\nabla F_n(\mathbf{x}) - \nabla f(\mathbf{x}))\right\|^2$$
$$= \left\|\sum_{n=1}^{N}\left[\frac{1}{P}\sum_{t=t_0}^{t_0+P-1} q_t^n - \mu_{t_0}\right](\nabla F_n(\mathbf{x}) - \nabla f(\mathbf{x}))\right\|^2, \quad (13)$$

where the last equality is due to (12). Since (13) is equal to the LHS of (7), we have the following.

**Proposition 4.2** (Regularized participation). *If $\frac{1}{P}\sum_{t=t_0}^{t_0+P-1} q_t^n - \mu_{t_0} = 0$ for some $\mu_{t_0}$ and all $n \in \{1,\ldots,N\}$, $t_0 \in \{0, P, 2P, \ldots\}$, then (7) holds with $\tilde{\delta}^2(P) = 0$.*

This implies that if $P$ is chosen so that the averaged participation weights over every interval of $P$ rounds $(\frac{1}{P}\sum_{t=t_0}^{t_0+P-1} q_t^n)$ are equal to each other among all the clients $n \in \{1,\ldots,N\}$, i.e., regularized, then $\tilde{\delta}^2(P) = 0$. Together with Corollary 3.2 and Proposition 4.1, we have the following.

**Corollary 4.3** (Regularized participation). *If the condition of Proposition 4.2 holds for $P \leq \frac{T}{2}$, and we choose the learning rates according to Corollary 3.2, then*

$$\min_t \mathbb{E}\left[\left\|\nabla f(\mathbf{x}_t)\right\|^2 \middle| \mathcal{Q}\right] \leq \mathcal{O}\left(\frac{\sigma\rho\sqrt{L\mathcal{F}}}{\sqrt{IT}} + \frac{LP\mathcal{F} + d^2 + \sigma^2}{T}\right). \tag{14}$$

If Proposition 4.2 holds for a finite $P$ that does not depend on $T$, we obtain a convergence rate of $\mathcal{O}\left(\frac{\sigma\rho}{\sqrt{IT}} + \frac{1+\sigma^2}{T}\right)$ for $T \geq 2P$, where all the other constants are ignored in the $\mathcal{O}(\cdot)$ notation here. For full gradient descent with $\sigma^2 = 0$, this convergence rate is improved to $\mathcal{O}\left(\frac{1}{T}\right)$.

***Interpreting $\tilde{\delta}^2(P)$ Using Variance.*** The condition for $\tilde{\delta}^2(P) = 0$ may be too stringent for some practical scenarios of FL. Next, we show that $\tilde{\delta}^2(P)$ can be expressed as the variance of client participation weights averaged over $P$ rounds. We first express $\tilde{\delta}^2(P)$ with $d^2$ by further bounding:

$$(13) \leq N\sum_{n=1}^{N}\left[\overline{q_{t_0}^n} - \mu_{t_0}\right]^2 d^2, \tag{15}$$

where we first use Jensen's inequality on the sum over $n$ in (13), then move the term $\overline{q_{t_0}^n} - \mu_{t_0}$ to the outside of the norm, where we define $\overline{q_{t_0}^n} := \frac{1}{P}\sum_{t=t_0}^{t_0+P-1} q_t^n$ for any $n \in \{1,\ldots,N\}$, and afterwards bound $\left\|\nabla F_n(\mathbf{x}) - \nabla f(\mathbf{x})\right\|^2$ by $d^2$ according to Assumption 3. We note that (7) holds if the RHS of (15) is upper bounded by $\tilde{\delta}^2(P)$. Therefore, we can obtain $\tilde{\delta}^2(P)$ by analyzing the statistical properties of $\overline{q_{t_0}^n} - \mu_{t_0}$ and then choosing $\tilde{\delta}^2(P)$ to be equal to the RHS of (15).

In the following, we assume that $\mu_{t_0}$ is chosen such that $\mu_{t_0} = \mathbb{E}[q_t^n]$ for all $n \in \{1,\ldots,N\}$ and $t \in \{t_0, t_0 + 1, \ldots, t_0 + P - 1\}$. This implies that the mean participation weights of all the clients within the same cycle of $P$ rounds are equal. Note that this condition is the same as unbiased client sampling in existing works [8, 9, 18, 21, 22, 36], because we consider an unweighted average in (1) and any weighting is included in the local objective function $F_n(\mathbf{x})$. However, differently from these existing works, we do not assume independence here (we will only assume independence in a special case later). Under this condition, it is apparent that $\text{Var}\left(\overline{q_{t_0}^n}\right) = \mathbb{E}\left[\left(\overline{q_{t_0}^n} - \mu_{t_0}\right)^2\right]$ is the variance of $q_t^n$ averaged over $P$ rounds, for any $n \in \{1,\ldots,N\}$. We immediately have the following result.

**Proposition 4.4** (Ergodic participation). *If $\{q_t^n : \forall t\}$ is a mean-ergodic process for any $n \in \{1,\ldots,N\}$, which means that $\lim_{P\to\infty}\mathbb{E}\left[\left(\overline{q_{t_0}^n} - \mu_{t_0}\right)^2\right] = 0$ for any $n$ and $t_0$, then there exists $\tilde{\delta}^2(P)$ such that $\lim_{P\to\infty}\tilde{\delta}^2(P) = 0$ while satisfying (7) in expectation.*

Next, we consider the class of stationary and strongly mixing processes. Informally, a random process is said to be *strongly mixing* with coefficient $\alpha(P)$ if the outcomes that are at least $P$ steps apart are nearly independent, where the distance between the joint probability distribution of the outcomes and their independent counterpart is at most $\alpha(P)$. See [2, Sec. 27] for a formal definition. A specific example of stationary and strongly mixing processes is finite-state irreducible and aperiodic Markov chains, which have an $\alpha(P)$ that exponentially decreases in $P$ [2, Thm. 8.9]. It is known that the following (generalized) central limit theorem holds for stationary and strongly mixing processes, where we adapt the result to our participation weights.

**Lemma 4.5** ([2, Thm. 27.4]). *If $\{q_t^n : \forall t\}$ is stationary and strongly mixing with $\alpha(P) = \mathcal{O}(P^{-5})$, for any $n \in \{1,\ldots,N\}$, then $\overline{q_{t_0}^n} \sim \mathcal{N}\left(\mu_{t_0}, \frac{\hat{v}^2}{P}\right)$ for any $t_0$ when $P \to \infty$, where $\mathcal{N}(\cdot,\cdot)$ denotes the normal distribution and $\hat{v}^2 := \text{Var}(q_{t_0}^n) + 2\sum_{p=1}^{\infty}\text{Cov}(q_{t_0}^n, q_{t_0+p}^n)$.*

From the definition of strongly mixing, we know that $\text{Cov}(q_{t_0}^n, q_{t_0+p}^n)$ in the definition of $\hat{v}^2$ approaches zero as $p$ gets large, hence $\sum_{p=1}^{\infty}\text{Cov}(q_{t_0}^n, q_{t_0+p}^n)$ converges to a finite value. When $P \to \infty$

and $\alpha(P) = \mathcal{O}(P^{-5})$, Lemma 4.5 shows that $\mathrm{Var}\left(\overline{q_{t_0}^n}\right) = \frac{\hat{\upsilon}^2}{P}$. Chebyshev's inequality gives

$$\Pr\left\{\left(\overline{q_{t_0}^n} - \mu_{t_0}\right)^2 \leq \frac{\hat{\upsilon}^2}{cP}\right\} \geq 1 - c. \tag{16}$$

Plugging $\mathrm{Var}\left(\overline{q_{t_0}^n}\right) = \mathbb{E}\left[\left(\overline{q_{t_0}^n} - \mu_{t_0}\right)^2\right] = \frac{\hat{\upsilon}^2}{P}$ and $\left(\overline{q_{t_0}^n} - \mu_{t_0}\right)^2 \leq \frac{\hat{\upsilon}^2}{cP}$ into the RHS of (15) gives the following expected and high-probability results, respectively, where "w.p." stands for "with probability".

**Proposition 4.6** (Stationary and strongly mixing participation). *If $\{q_t^n : \forall t\}$ is stationary and strongly mixing with $\alpha(P) = \mathcal{O}(P^{-5})$, for any $n \in \{1, \ldots, N\}$, then as $P \to \infty$: Choosing $\tilde{\delta}^2(P) = \frac{N^2 d^2 \hat{\upsilon}^2}{P}$ satisfies (7) in expectation; $\tilde{\delta}^2(P) = \frac{N^2 d^2 \hat{\upsilon}^2}{cP}$ satisfies (7) w.p. $1 - c$.*

For independent participation, we can obtain a similar result while *not* requiring $P \to \infty$. Assuming that $\mathrm{Var}(q_t^n) \leq \upsilon^2$ for any $t$ and $n$, it is evident that $\mathrm{Var}\left(\overline{q_{t_0}^n}\right) \leq \frac{\upsilon^2}{P}$, because the variance of the sum of independent random variables is equal to the sum of the variance, and $\mathrm{Var}(aZ) = a^2\mathrm{Var}(Z)$, for an arbitrary constant $a$ and random variable $Z$. We note that Hoeffding's inequality [16] gives

$$\Pr\left\{\left(\overline{q_{t_0}^n} - \mu_{t_0}\right)^2 \leq \frac{\ln(2/c)}{2P}\right\} \geq 1 - c. \tag{17}$$

We have the following result by plugging into the RHS of (15).

**Proposition 4.7** (Independent participation). *If $\{q_t^n : \forall t\}$ is independent across $t$, for any $n$, then: Choosing $\tilde{\delta}^2(P) = \frac{N^2 d^2 \upsilon^2}{P}$ satisfies (7) in expectation; $\tilde{\delta}^2(P) = \frac{N^2 d^2 \ln(2/c)}{2P}$ satisfies (7) w.p. $1 - c$.*

Note that the independence here is only assumed across $t$, so $q_t^n$ and $q_t^{n'}$ for the same $t$ but $n \neq n'$ can still be dependent of each other. Compared to Proposition 4.6, the high-probability bound in Proposition 4.7 has $c$ in the logarithmic term, which is tighter since $\frac{1}{2}\ln\left(\frac{2}{c}\right) \leq \frac{1}{c} - \frac{1}{2}$ (ignoring $\hat{\upsilon}^2$).

***Choosing $P$ as a Function of $T$.*** The choices of $\tilde{\delta}^2(P)$ in Propositions 4.6 and 4.7 include $P$ in the denominator. Hence, we can guarantee convergence to zero error if we choose $P$ to be an increasing function of $T$, which also ensures that $P \to \infty$ as $T \to \infty$. If we choose $P \propto N^{\frac{5}{2}}\sqrt{IT}$, we can obtain the following convergence rate from Corollary 3.3 together with Proposition 4.1.

**Corollary 4.8** (Stochastic participation). *If the conditions of either Proposition 4.6 or Proposition 4.7 hold, and we choose $P = \hat{\upsilon}^2 N^{\frac{5}{2}}\sqrt{IT}$ or $P = \upsilon^2 N^{\frac{5}{2}}\sqrt{IT}$, respectively, and the learning rates according to Corollary 3.3, then the convergence error in expectation over $\mathcal{Q}$ satisfies*

$$\min_t \mathbb{E}\left[\|\nabla f(\mathbf{x}_t)\|^2\right] \leq \mathcal{O}\left(\frac{(1 + \sigma^2)\rho\sqrt{L\mathcal{F}}}{\sqrt{IT}} + \frac{d^2}{\sqrt{NIT}} + \frac{d^2 + \sigma^2}{T}\right), \tag{18}$$

*for $I \leq \frac{\rho}{\hat{\upsilon}^2 N^{5/2}\sqrt{L\mathcal{F}}}$ and $T \to \infty$ when using conditions of Proposition 4.6, or $I \leq \frac{\rho}{\upsilon^2 N^{5/2}\sqrt{L\mathcal{F}}}$ and $T \geq 4\upsilon^4 IN^5$ when using condition of Proposition 4.7.*

**Remark.** Although the upper bound of $I$ in Corollary 4.8 may appear restrictive, note that the optimal solution $\mathbf{x}^*$ to (1) remains the same when the objective $f(\mathbf{x})$ is scaled by a multiplicative positive constant, which in turn scales $\sqrt{L\mathcal{F}}$. More specifically, consider an arbitrary $a > 0$, we define $F'(\mathbf{x}) := aF(\mathbf{x})$ and $f'(\mathbf{x}) := \frac{1}{N}\sum_{n=1}^N F_n'(\mathbf{x}) = af(\mathbf{x})$. Then, we have $f'(\mathbf{x}_0) - f'^* = af(\mathbf{x}_0) - af^* = a\mathcal{F}$. We also have $\|\nabla F_n'(\mathbf{x}) - \nabla F_n'(\mathbf{y})\| = a\|\nabla F_n(\mathbf{x}) - \nabla F_n(\mathbf{y})\| \leq aL\|\mathbf{x} - \mathbf{y}\|$ from Assumption 1, due to the linearity of gradients. Hence, by choosing $a \in (0, 1)$ and replacing $F(\mathbf{x})$ with $F'(\mathbf{x})$, we can make $\sqrt{L\mathcal{F}}$ arbitrarily small and $\frac{\rho}{\hat{\upsilon}^2 N^{5/2}\sqrt{L\mathcal{F}}}$ or $\frac{\rho}{\upsilon^2 N^{5/2}\sqrt{L\mathcal{F}}}$ (the upper bound of $I$ in Corollary 4.8) arbitrarily large without affecting the optimal solution $\mathbf{x}^*$. Thus, we can potentially allow arbitrarily large $I$ by scaling $f(\mathbf{x})$ without changing the optimal solution $\mathbf{x}^*$. We leave the in-depth study of this phenomenon for future work, where we recognize that such a scaling will affect the LHS of the convergence bound too, although the optimal solution $\mathbf{x}^*$ does not change.

We also give the following further insight related to Corollary 4.8. In Corollary 4.8, we choose $P$ to be proportional to the variance $\hat{\upsilon}^2$ or $\upsilon^2$. This is intuitive because when the clients' participation weights have higher variance, we would like to wait for more rounds before amplifying, so that the contributions of clients are more balanced. In the same way, $P$ increases with the number of clients $N$. Recall that $\sum_{n=1}^N q_t^n = 1$. Intuitively, when the participation patterns of individual clients remain

"unchanged", $q_t^n$ scales with $\frac{1}{N}$ and its variance $\hat{v}^2$ or $v^2$ scales with $\frac{1}{N^2}$, so the product $\hat{v}^2 N^{\frac{5}{2}} \sim \sqrt{N}$ or $v^2 N^{\frac{5}{2}} \sim \sqrt{N}$, implying that $P$ effectively scales with $\sqrt{N}$ when this intuition holds.

While the result in Corollary 4.8 is in expectation, the high-probability bound is similar (see Table 1). In general, since (7) is used as an upper bound in the proof of Theorem 3.1, if $\tilde{\delta}^2(P)$ is chosen so that (7) holds in expectation or with a certain probability, such as in Propositions 4.4, 4.6 and 4.7, then the final convergence result also holds in expectation or with a certain probability, respectively.

## 5  Discussions and Insights

**Interpretation of "Linear Speedup".** Linear speedup is a desirable property seen in existing works that consider idealized client participation [36, 39, 40]. It essence, it means that the same convergence error can be achieved by increasing the number of participating clients $S$ (with $S \leq N$) and reducing the number of rounds $T$, while keeping the product $ST$ unchanged. In our case of arbitrary client participation, the coefficient $\rho$ is a generalization of the $1/\sqrt{S}$ term for linear speedup in existing works that select a fixed number of $S$ clients in each round. When $S$ clients participate with equal weight, because $\sum_{n=1}^{N} q_t^n = 1$, we have $q_t^n = 1/S$ for the participating clients and $q_t^n = 0$ for the non-participating clients, so $\rho = \left[ \sum_{n=1}^{N} (q_t^n)^2 \right]^{1/2} = 1/\sqrt{S}$. Plugging this $\rho$ back to the convergence results, we achieve a convergence rate of $\mathcal{O}\left(\sigma/\sqrt{SIT}\right)$ in Corollary 4.3 and $\mathcal{O}\left((1+\sigma^2)/\sqrt{SIT}\right)$ in Corollary 4.8, for sufficiently large $T$ while ignoring the other variables in $\mathcal{O}(\cdot)$, as shown in Table 1. This shows that we can achieve linear speedup in $S$ in this special case. We can also generalize to having at least $S_{\min}$ clients participating with equal weight in each round. Following the same argument, we have $\rho \leq 1/\sqrt{S_{\min}}$ in this case, giving a linear speedup in $S_{\min}$. In general, we achieve a linear speedup factor of $1/\rho^2$ in the case of arbitrary participation.

**Matching Lower Bound or State-of-the-Art Results.** We continue to assume sufficiently large $T$. For regularized participation (Corollary 4.3), our convergence rate of $\mathcal{O}\left(\sigma/\sqrt{SIT}\right)$ *matches the lower bound of convergence error for centralized SGD* [1]. The lower bound states that to reach a convergence error of $\epsilon$, there exists an objective function such that at least $\Omega\left(\sigma^2/\epsilon^2\right)$ stochastic gradient oracle calls are required.[2] We have $SIT$ oracle calls to reach $\epsilon = \mathcal{O}\left(\sigma/\sqrt{SIT}\right)$ according to our result, thus our upper bound matches this lower bound and is asymptotically optimal. Note that our upper bound still matches the lower bound if we include the coefficient $\sqrt{L\mathcal{F}}$ in $\mathcal{O}(\cdot)$ and $L\mathcal{F}$ in $\Omega(\cdot)$. For stochastic participation (Corollary 4.8), our convergence rate of $\mathcal{O}\left((1+\sigma^2)/\sqrt{SIT}\right)$ *matches state-of-the-art results of FedAvg* [8, 18, 36] that were obtained for the more idealized case of clients being selected to participate according to a specific independent sampling scheme.

**Special Case of Waiting for All Clients.** We consider a specific configuration of FedAvg where all the clients wait for $P$ rounds before proceeding with SGD. In every iteration $i$ of round $t$ within these $P$ rounds, each participating client $n$ may call the stochastic gradient oracle to obtain $\mathbf{g}_n(\mathbf{x}_t)$, but it does not perform local updates. Instead, it accumulates and averages the sampled instances of $\mathbf{g}_n(\mathbf{x}_t)$. SGD is performed once after these $P$ rounds using the *average* $\mathbf{g}_n(\mathbf{x}_t)$ from each client $n$. Assume that $P$ is large enough so that each client $n$ computes $\mathbf{g}_n(\mathbf{x}_t)$ at least $M \geq 1$ times that are averaged afterwards. If the $M$ instances of $\mathbf{g}_n(\mathbf{x}_t)$ are all independent from each other, the noise $\sigma^2$ (defined in Assumption 2) becomes $\sigma^2/M$. This configuration can be considered as a special case of Algorithm 1, where all $N$ clients perform SGD once in $P$ rounds, so there are $T/P$ updates in total, with a reduced stochastic gradient noise of $\sigma^2/M$. By replacing $\sigma$ and $SIT$ in our bound $\mathcal{O}\left(\sigma/\sqrt{SIT}\right)$ with $\sigma/\sqrt{M}$ and $NT/P$, respectively, we obtain a convergence rate of $\mathcal{O}\left(\sigma\sqrt{P}/\sqrt{MNT}\right)$ for regularized participation with this "wait-for-all" method.

In theory, when considering regularized participation and $M$ is large enough, the convergence rate of this "wait-for-all" method can potentially match with that of the original configuration of Algorithm 1. In this way, the convergence error upper bound of both methods can match the lower bound of SGD convergence error (see discussion above). However, in practice, reducing the stochastic gradient noise as in this "wait-for-all" method may only give limited improvement. One reason is that practical settings have a finite number of training data samples, so the noise can only be reduced by a certain degree. In addition, having some noise in SGD can prevent the model parameter from being trapped in saddle points and local minima in practice. The empirical results in Section 6 show that Algorithm 1

---

[2]Note that [1] defines the convergence error as $\|\nabla f(\mathbf{x}_t)\|$ whereas we define it as $\|\nabla f(\mathbf{x}_t)\|^2$. Hence, the order of our $\epsilon$ is different from that in [1].

with amplification performs better than waiting for all the clients. We also note that our results for stochastic participation (Corollary 4.8) *does not* require all the clients to participate within $P$ rounds.

**Achieving Regularized Participation in Practice.** In theory, regularized participation requires that $\frac{1}{P} \sum_{t=t_0}^{t_0+P-1} q_t^n$ is equal to each other for all the clients $n \in \{1, \ldots, N\}$, for a properly chosen $P$. When all the clients are connected to the system and client sampling is performed to limit the computation and communication overhead, this condition holds when the clients are selected according to a permutation, i.e., all the clients participate once before the same client can be selected again. In the case where clients get disconnected from time to time, the system can try to properly schedule the participation of connected clients and their weights $\{q_t^n\}$, so that within a cycle of $P$ rounds, all the clients participate with equal averaged weights. In practice, it can be sufficient if this regularized condition is only approximately satisfied, especially when multiple clients have similar datasets so that $\tilde{\delta}^2(P)$ is small although not necessarily zero. The empirical results in Section 6 show that Algorithm 1 with amplification can give good performance even if $P$ is less than the participation cycle of all clients. This suggests that, as long as the subset of clients that participate in $P$ rounds are more representative of the overall data distribution than the (usually much smaller) subset of clients that participate in a single round, amplifying the updates every $P$ rounds can be useful.

## 6 Experiments

We ran experiments of training convolutional neural networks (CNNs) with FashionMNIST [34] and CIFAR-10 [19] datasets, each of which has images in 10 different classes. We set the total number of clients to $N = 250$. Similar to existing works [6, 7, 36], we partition the data into clients so that each client has data of one majority class label, with $5\%$ of data of other (minority) labels, to simulate a setup with non-IID data distribution which is often encountered in FL scenarios. We assume that the clients' availability exhibits a periodic pattern inspired by [6, 7]. Namely, in the first 100 rounds, only clients with the first two majority labels are available; in the next 100 rounds, only clients with the next two majority labels are available, and so on. In each round, $S = 10$ clients participate in a regularized manner, out of all the currently available clients. To speed up initial rounds of training, similar to practical deep learning implementations, we start with standard FedAvg with a relatively large initial learning rate. The initial rates are $\gamma = 0.1$ and $\gamma = 0.05$ without amplification (i.e., $\eta = 1$) for FashionMNIST and CIFAR-10, respectively, which were obtained using grid search in a separate scenario of always participation. After an initial training of $2,000$ rounds for FashionMNIST and $4,000$ rounds for CIFAR-10, we study the performance of different approaches with their own learning rates. Additional setup details and results are given in Appendix D.

**Comparing Different Methods.** We show the results of Algorithm 1 both with and without amplification. When using amplification, we set $\eta = 10$ and $P = 500$. We also compare to an algorithm that waits for all the clients, related to our discussion in Section 5, for two settings where each client computes its gradients either on a minibatch or on the entire dataset, which we refer to as "wait-minibatch" and "wait-full", respectively. The best learning rate $\gamma$ of each approach was separately found on a grid that is $\{1, 0.1, 0.01, 0.001, 0.0001\}$ times the initial learning rate, with the periodic participation pattern described above. The results are shown in Figure 1.

We make a few key observations as follows. First, we clearly see that *Algorithm 1 with amplification gives the best performance*. By choosing $P = 500$, we match with the participation period of clients, as described in the setup above. This corresponds to our setting with regularized participation. In

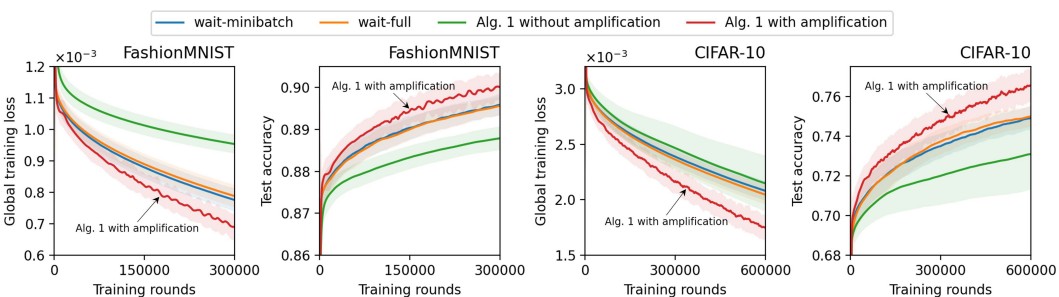

Figure 1: Results for different approaches with periodically connected clients ($P = 500$).

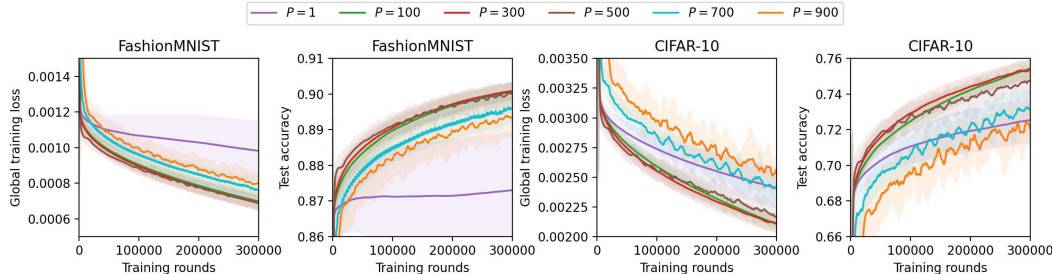

Figure 2: Algorithm 1 with amplification and different $P$, with periodically connected clients.

this case, Algorithm 1 with amplification allows different groups of clients to make small progress among themselves first. After $P$ rounds, each client has made its own share of contribution, and the updates get amplified so that the model parameter progresses faster towards the direction that is overall beneficial for all the clients. As our theory predicts, this approach gives a desirable convergence rate, which is now verified by experiments. Next, we observe that *the "wait-minibatch" and "wait-full" methods perform worse than Algorithm 1 with amplification*. We also observe that "wait-full", which computes the gradient on the entire dataset, does not provide substantial gain compared to "wait-minibatch" in the experiments. These observations align with our discussion in Section 5 on the practical performance of different methods. Finally, we observe that *standard FedAvg (i.e., Algorithm 1 without amplification) gives the lowest performance*. This is because without amplification, the algorithm cannot put more emphasis on the collective updates by the cohort of all clients, and the parameter updates may diverge from the overall optimal direction.

**Different Values of $P$.** Next, we fix $\eta = 10$ in Algorithm 1 and study the effect of choosing different values of $P$. We use the same learning rate $\gamma$ that is obtained from grid search in the previous experiment. Note that the clients' availability pattern remains the same as described above. The choice of different $P$ simulates the practical scenario where the estimation of $P$ may not perfectly align with the actual participation cycle. The results are shown in Figure 2.

We observe that $P = 1$, which corresponds to the "classical" setting of FedAvg with two-sided learning rates [18, 36], does not give the best performance. Interestingly, compared to $P = 500$, we see that $P = 100$ and $P = 300$ give a similar performance, and even slightly better performance in the case of CIFAR-10 data. Due to the random offset applied to the first participation cycle in each experiment (see Appendix D.1 for details), every interval of 100 rounds can include the "partial" contributions by two subsets of clients (with data in 4, out of 10, majority classes). The results suggest that amplifying such partial contributions by multiple subsets of clients can still improve performance. In the case of CIFAR-10 data, the reason for $P = 500$ being slightly worse may be that the accumulated update within $P$ rounds generally becomes larger as $P$ gets larger, in which case amplification causes a bigger change in the model parameter. Depending on the landscape of the loss function, this change may be too big so that the model performance decreases. For a similar reason, the performances of $P = 700$ and $P = 900$ are even worse. Nevertheless, we expect that choosing a smaller learning rate $\gamma$ can improve the performance for large $P$ values.

## 7   Conclusion

In this paper, we have studied FL with arbitrary client participation. For a generalized FedAvg algorithm that amplifies parameter updates every $P$ rounds, we have developed a unified framework for convergence analysis and obtained convergence rates for a variety of client participation patterns. Our findings suggest that regularized participation with finite $P$ gives the best performance and matches the lower bound of SGD convergence error when $T$ is sufficiently large. For stochastic participation, first, we have formally proven that convergence is guaranteed when the participation process is ergodic. Then, for two generic classes of participation processes, we have proven that with a properly chosen $P$, our convergence rate matches state-of-the-art FedAvg convergence rates that were derived for the idealized case of independent and unbiased participation. The empirical results have confirmed that amplification is useful and also provided further insights. Future directions include analysis of advanced FL algorithms and more detailed empirical study.

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
