# Appendix

# A  Additional Related Works

We complement Section 1 by discussing a few additional related works as follows.

Client unavailability was also studied from the perspective of block-cyclic SGD [6, 7], which trains *multiple* models each for a block (group) of clients with known identity and was analyzed for convex objectives. Different from those works, our goal is to train a *single* model without requiring block structure, and we focus on non-convex objectives that frequently arise in modern problems involving neural networks.

In the literature of SGD in the centralized setting, the effect of different ways of minibatch sampling has also been studied, such as IID sampling from an arbitrary distribution [11] and random reshuffling [13, 25]. However, the problem in FL is much more complex and needs to incorporate the diversity of different clients' local objectives. Moreover, we focus on a unified framework that is more general than the special cases of IID sampling and random reshuffling.

FedAvg with two-sided learning rates has been studied in the literature [18, 36], where a global learning rate is applied to the updates at the end of each round, giving an improved convergence bound compared to using a single-sided learning rate. This is equivalent to our Algorithm 1 with $P = 1$ and the amplification factor $\eta$ being the global learning rate. We extend this setup by allowing any $P \geq 1$. Moreover, most existing works including [18, 36] have only shown the benefit of two-sided learning rates in a theoretical setting; their experiments still uses a global learning rate of $1.0$, which is equivalent to the case of a single-sided learning rate. Although some recent works have used different global learning rates in experiments, the best setting often remains to be choosing $1.0$ as the global learning rate in the case of FedAvg [26, Table 8]. In contrast, we have verified the benefit of amplification in our experiments (Section 6), for scenarios with more challenging client participation patterns than existing works. The idea of amplification is also related to look-ahead algorithms [31, 42], which consider a two-agent setting, e.g., a server and a single client. Different from these works, we consider the unique challenges in federated learning with multiple arbitrarily participating clients, where the data is non-IID across clients.

It is also worth pointing out that our discussion related to waiting for all clients in Section 5 has some analogy to the comparison between local SGD and minibatch SGD [32, 33, 41]. Even when not considering the effect of partial participation, these existing works have only identified some specific classes of convex objectives where local SGD can be shown to have a better theoretical convergence rate than minibatch SGD. An improved theoretical approach to capture the fact that local SGD often outperforms minibatch SGD in practice, for a wide range of problems and objective functions (especially non-convex objectives), remains an interesting future direction.

# B  Additional Discussions

## B.1  Why Does Amplification Help?

We give an illustrative example to explain why the use of amplification with $\eta > 1$ generally improves the performance, as shown in both our theory and experiments. Consider a simple setting with $N = 3$ clients. Each client $n$ has a local objective defined as $F_n(\mathbf{x}) = \frac{1}{2} \|\mathbf{x} - \mathbf{z}_n\|^2$ for some constant vector $\mathbf{z}_n \in \mathbb{R}^m$. It is evident that the true optimal solution that minimizes the global objective $f(\mathbf{x}) = \frac{1}{6} \sum_{n=1}^{3} \|\mathbf{x} - \mathbf{z}_n\|^2$ is $\mathbf{x}^* = \frac{1}{3} \sum_{n=1}^{3} \mathbf{z}_n$, but we assume that this is not known to the system and we would like to solve this problem using Algorithm 1. In this specific example, we consider a two-dimensional space (i.e., $m = 2$) and define $\mathbf{z}_1 = (-1, 0), \mathbf{z}_2 = (1, 0), \mathbf{z}_3 = (0, \sqrt{3})$. We further assume that the three clients are available in a cyclic manner, so that in the first round, only client $n = 1$ is available; in the second round, only client $n = 2$ is available; in the third round, only client $n = 3$ is available; then this cycle continues with client $n = 1$ available in the fourth round, and so on. We plot the trajectory of how the model parameter $\mathbf{x}$ changes from the initial value $\mathbf{x}_0$ to $\mathbf{x}_{15}$ that is obtained after $T = 15$ rounds. The initial parameter is set to $\mathbf{x}_0 = (1, 2)$ in this example. The plots of the trajectories with different choices of local learning rate $\gamma$ and amplification factor $\eta$ are shown in Figures B.1–B.2.

With the local objectives defined in this example, the local gradient can be directly computed as $\nabla F_n(\mathbf{x}) = \mathbf{x} - \mathbf{z}_n$. Therefore, in the case of no amplification (i.e., $\eta = 1$), the solution variable $\mathbf{x}$ moves towards $\mathbf{z}_1$ in the first round when client $n = 1$ is available; then, it moves towards $\mathbf{z}_2$ in the

second round when client $n = 2$ is available, and so on. As we see from Figure B.1, this can lead to a pattern where $\mathbf{x}$ cycles around the optimal solution $\mathbf{x}^*$ but approaches $\mathbf{x}^*$ only very slowly, especially when $\gamma$ is large. In the case of small $\gamma$, this cyclic pattern is not apparent, but we can still see that $\mathbf{x}$ moves towards different local optimal points (i.e., $\mathbf{z}_n$ for different $n$) in different rounds, and the overall convergence speed is slow. This leads to the following problem: regardless of whether we choose a large or small $\gamma$, at the end of the 15-th round, there is an apparent gap between $\mathbf{x}_{15}$ and $\mathbf{x}^*$.

The advantage of amplification is that it increases the importance of the aggregated updates made by clients that participate in different rounds. In this specific example, a single client's update direction may be towards its own local optimum, which can be different from the direction towards the global optimum. However, the accumulated updates by all the three clients over every three rounds will be more likely towards the global optimum (or a neighborhood of it). By amplifying such accumulated updates after every three rounds (i.e., $P = 3$), we can let the solution variable $\mathbf{x}$ move much faster towards the true (global) optimum. This intuition is confirmed by the trajectories shown in Figure B.2. Here, we note that the product of $\gamma$ and $\eta$ in the corresponding sub-figures of Figures B.1 and B.2 are equal. By keeping a small local learning rate $\gamma$ and amplifying the updates accumulated in every $P = 3$ rounds, we can converge to the optimal value $\mathbf{x}^*$, as shown in Figure B.2.

Note that in practice, $P$ does not need to perfectly align with the "cycle" of participation, as seen in Figure 2 in Section 6 of the main paper.

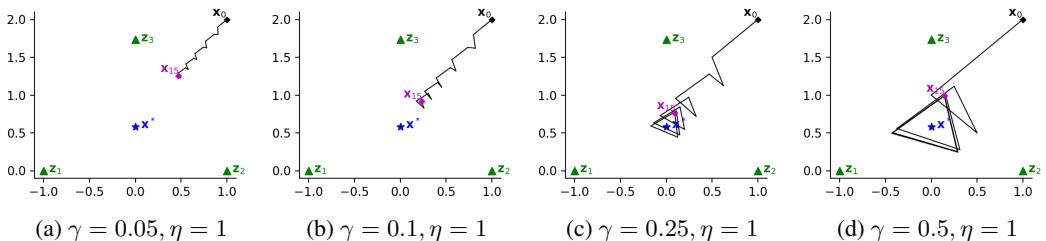

(a) $\gamma = 0.05, \eta = 1$     (b) $\gamma = 0.1, \eta = 1$     (c) $\gamma = 0.25, \eta = 1$     (d) $\gamma = 0.5, \eta = 1$

Figure B.1: Motivating example: different local learning rates $\gamma$ without amplification (i.e., $\eta = 1$). The trajectory from $\mathbf{x}_0$ to $\mathbf{x}_{15}$ shows how the model parameter changes from round $t = 0$ to round $t = 15$.

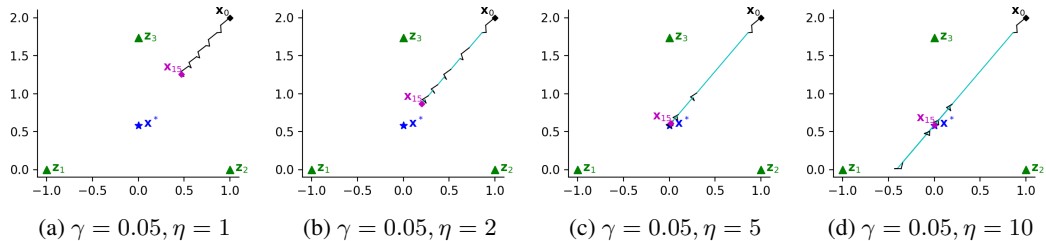

(a) $\gamma = 0.05, \eta = 1$     (b) $\gamma = 0.05, \eta = 2$     (c) $\gamma = 0.05, \eta = 5$     (d) $\gamma = 0.05, \eta = 10$

Figure B.2: Motivating example: fixed local learning rate $\gamma = 0.05$ with different amplification factors $\eta$. The trajectory from $\mathbf{x}_0$ to $\mathbf{x}_{15}$ shows how the model parameter changes from round $t = 0$ to round $t = 15$. The segments in cyan color shows the change in parameter $\mathbf{x}$ due to amplification, while the segments in black color shows the parameter change due to regular SGD operation.

## B.2 Relating Different Participation Patterns to Practical FL Scenarios

In the following, we discuss some possible connections between practical FL application scenarios and the different participation patterns that are considered in our theoretical analysis.

The case of periodic shifts, which is often seen in practical cross-device FL scenarios, is mostly related to *regularized participation*. In this case, the clients participate equally over the entire period, but only a subset of clients from a specific population (and with a specific data distribution) may participate in each "period" that includes a certain number of consecutive rounds. In practice, this "equal participation" only needs to hold approximately, and our empirical results at the end of

Section 6 show that our Algorithm 1 can still give good performance even if $P$ is less than the full cycle (period) of participation.

Among the stochastic participation patterns, *ergodic* is the most generic. Intuitively, it says that the participation weights of different clients are the same when averaged over a long enough time. This can represent a cross-device FL scenario, where the participation of each client at any time instance can be highly random, but the long-term average statistics of clients are the same. The case of *strongly mixing* participation includes the special case where the participation follows a Markov chain, i.e., if the client is currently unavailable, there is a certain (possibly small) probability that it will become available in the next round, and vice versa. This can also represent a cross-device FL setting, where the devices get connected and disconnected over time, and there is some randomness in the exact time it gets connected or disconnected. Finally, the case of *independent* participation may represent a cross-silo FL setting, where the reason for not participating in a certain round is not because the client is disconnected for an extended period of time, but rather because the client has some other higher-priority tasks to run so that it cannot participate in all rounds.

## C  Proofs

### C.1  Preliminaries

The following inequalities are frequently used throughout our proofs. We will use them without further explanation.

From Jensen's inequality, for any $\mathbf{z}_m \in \mathbb{R}^d, m \in \{1, 2, \ldots, M\}$, we have

$$\left\| \frac{1}{M} \sum_{m=1}^{M} \mathbf{z}_m \right\|^2 \leq \frac{1}{M} \sum_{m=1}^{M} \|\mathbf{z}_m\|^2 \tag{C.1}$$

which directly gives

$$\left\| \sum_{m=1}^{M} \mathbf{z}_m \right\|^2 \leq M \sum_{m=1}^{M} \|\mathbf{z}_m\|^2 . \tag{C.2}$$

Peter-Paul inequality (also known as Young's inequality) gives

$$\langle \mathbf{z}_1, \mathbf{z}_2 \rangle \leq \frac{b \|\mathbf{z}_1\|^2}{2} + \frac{\|\mathbf{z}_2\|^2}{2b} \tag{C.3}$$

for any $b > 0$ and any $\mathbf{z}_1, \mathbf{z}_2 \in \mathbb{R}^d$. Hence, we also have

$$\|\mathbf{z}_1 + \mathbf{z}_2\|^2 \leq (1 + b) \|\mathbf{z}_1\|^2 + \left(1 + \frac{1}{b}\right) \|\mathbf{z}_2\|^2 . \tag{C.4}$$

Throughout the analysis, we use the short-hand notation $\mathbb{E}_{\cdot|\mathcal{Q}}[\cdot]$ to denote $\mathbb{E}[\cdot|\mathcal{Q}]$.

In addition, we use the notations in Algorithm 1 and Assumptions 1, 2, and 3'.

### C.2  Lemmas

**Lemma C.1.** *When $\gamma \leq \frac{1}{12LIP}$, we have*

$$\sum_{n=1}^{N} q_t^n \mathbb{E}_{\cdot|\mathcal{Q}} \left[ \left\| \mathbf{y}_{t,i}^n - \sum_{n'=1}^{N} q_t^{n'} \mathbf{y}_{t,i}^{n'} \right\|^2 \right] \leq 3I\gamma^2 \left( 6I\tilde{\nu}^2 + \sigma^2 \left( 1 - \sum_{n=1}^{N} (q_t^n)^2 \right) \right). \tag{C.5}$$

*Proof.* Note that

$$\sum_{n=1}^{N} q_t^n \mathbb{E}_{\cdot|\mathcal{Q}} \left[ \left\| \mathbf{y}_{t,i+1}^n - \sum_{n'=1}^{N} q_t^{n'} \mathbf{y}_{t,i+1}^{n'} \right\|^2 \right]$$

$$= \sum_{n=1}^{N} q_t^n \mathbb{E}_{\cdot|\mathcal{Q}} \left[ \left\| \mathbf{y}_{t,i}^n - \gamma \mathbf{g}_n(\mathbf{y}_{t,i}^n) - \sum_{n'=1}^{N} q_t^{n'} (\mathbf{y}_{t,i}^{n'} - \gamma \mathbf{g}_{n'}(\mathbf{y}_{t,i}^{n'})) \right\|^2 \right]$$

$$\overset{(a)}{=} \sum_{n=1}^{N} q_t^n \mathbb{E}_{\cdot|\mathcal{Q}} \left[ \left\| \mathbf{y}_{t,i}^n - \gamma \nabla F_n(\mathbf{y}_{t,i}^n) - \sum_{n'=1}^{N} q_t^{n'}(\mathbf{y}_{t,i}^{n'} - \gamma \nabla F_{n'}(\mathbf{y}_{t,i}^{n'})) \right\|^2 \right]$$

$$+ \sum_{n=1}^{N} q_t^n \mathbb{E}_{\cdot|\mathcal{Q}} \left[ \left\| \gamma \left( \mathbf{g}_n(\mathbf{y}_{t,i}^n) - \nabla F_n(\mathbf{y}_{t,i}^n) \right) - \gamma \sum_{n'=1}^{N} q_t^{n'} \left( \mathbf{g}_{n'}(\mathbf{y}_{t,i}^{n'}) - \nabla F_{n'}(\mathbf{y}_{t,i}^{n'}) \right) \right\|^2 \right]$$

$$= \sum_{n=1}^{N} q_t^n \mathbb{E}_{\cdot|\mathcal{Q}} \left[ \left\| \mathbf{y}_{t,i}^n - \gamma \nabla F_n(\mathbf{y}_{t,i}^n) - \sum_{n'=1}^{N} q_t^{n'}(\mathbf{y}_{t,i}^{n'} - \gamma \nabla F_{n'}(\mathbf{y}_{t,i}^{n'})) \right\|^2 \right]$$

$$+ \sum_{n=1}^{N} q_t^n \mathbb{E}_{\cdot|\mathcal{Q}} \left[ \left\| - \gamma \sum_{n' \in \{1,\ldots,N\} \setminus \{n\}} q_t^{n'} \left( \mathbf{g}_{n'}(\mathbf{y}_{t,i}^{n'}) - \nabla F_{n'}(\mathbf{y}_{t,i}^{n'}) \right) \right. \right.$$

$$\left. \left. - \gamma \left( q_t^n - 1 \right) \left( \mathbf{g}_n(\mathbf{y}_{t,i}^n) - \nabla F_n(\mathbf{y}_{t,i}^n) \right) \right\|^2 \right]$$

$$\overset{(b)}{\leq} \sum_{n=1}^{N} q_t^n \mathbb{E}_{\cdot|\mathcal{Q}} \left[ \left\| \mathbf{y}_{t,i}^n - \gamma \nabla F_n(\mathbf{y}_{t,i}^n) - \sum_{n'=1}^{N} q_t^{n'}(\mathbf{y}_{t,i}^{n'} - \gamma \nabla F_{n'}(\mathbf{y}_{t,i}^{n'})) \right\|^2 \right] + \gamma^2 \sigma^2 \left( 1 - \sum_{n=1}^{N} (q_t^n)^2 \right)$$

$$= \sum_{n=1}^{N} q_t^n \mathbb{E}_{\cdot|\mathcal{Q}} \left[ \left\| \mathbf{y}_{t,i}^n - \sum_{n'=1}^{N} q_t^{n'} \mathbf{y}_{t,i}^{n'} - \gamma \left[ \nabla F_n(\mathbf{y}_{t,i}^n) - \nabla F_n \left( \sum_{n''=1}^{N} q_t^{n''} \mathbf{y}_{t,i}^{n''} \right) \right. \right. \right.$$

$$+ \nabla F_n \left( \sum_{n''=1}^{N} q_t^{n''} \mathbf{y}_{t,i}^{n''} \right) - \sum_{n'=1}^{N} q_t^{n'} \nabla F_{n'} \left( \sum_{n''=1}^{N} q_t^{n''} \mathbf{y}_{t,i}^{n''} \right)$$

$$\left. \left. \left. + \sum_{n'=1}^{N} q_t^{n'} \nabla F_{n'} \left( \sum_{n''=1}^{N} q_t^{n''} \mathbf{y}_{t,i}^{n''} \right) - \sum_{n'=1}^{N} q_t^{n'} \nabla F_{n'}(\mathbf{y}_{t,i}^{n'}) \right] \right\|^2 \right]$$

$$+ \gamma^2 \sigma^2 \left( 1 - \sum_{n=1}^{N} (q_t^n)^2 \right)$$

$$\leq \left( 1 + \frac{1}{2I-1} \right) \sum_{n=1}^{N} q_t^n \mathbb{E}_{\cdot|\mathcal{Q}} \left[ \left\| \mathbf{y}_{t,i}^n - \sum_{n'=1}^{N} q_t^{n'} \mathbf{y}_{t,i}^{n'} \right\|^2 \right]$$

$$+ 2I \gamma^2 \sum_{n=1}^{N} q_t^n \mathbb{E}_{\cdot|\mathcal{Q}} \left[ \left\| \nabla F_n(\mathbf{y}_{t,i}^n) - \nabla F_n \left( \sum_{n''=1}^{N} q_t^{n''} \mathbf{y}_{t,i}^{n''} \right) + \nabla F_n \left( \sum_{n''=1}^{N} q_t^{n''} \mathbf{y}_{t,i}^{n''} \right) \right. \right.$$

$$- \sum_{n'=1}^{N} q_t^{n'} \nabla F_{n'} \left( \sum_{n''=1}^{N} q_t^{n''} \mathbf{y}_{t,i}^{n''} \right) + \sum_{n'=1}^{N} q_t^{n'} \nabla F_{n'} \left( \sum_{n''=1}^{N} q_t^{n''} \mathbf{y}_{t,i}^{n''} \right)$$

$$\left. \left. - \sum_{n'=1}^{N} q_t^{n'} \nabla F_{n'}(\mathbf{y}_{t,i}^{n'}) \right\|^2 \right]$$

$$+ \gamma^2 \sigma^2 \left( 1 - \sum_{n=1}^{N} (q_t^n)^2 \right)$$

$$\leq \left( 1 + \frac{1}{2I-1} \right) \sum_{n=1}^{N} q_t^n \mathbb{E}_{\cdot|\mathcal{Q}} \left[ \left\| \mathbf{y}_{t,i}^n - \sum_{n'=1}^{N} q_t^{n'} \mathbf{y}_{t,i}^{n'} \right\|^2 \right]$$

$$+ 6I \gamma^2 \sum_{n=1}^{N} q_t^n \mathbb{E}_{\cdot|\mathcal{Q}} \left[ \left\| \nabla F_n(\mathbf{y}_{t,i}^n) - \nabla F_n \left( \sum_{n''=1}^{N} q_t^{n''} \mathbf{y}_{t,i}^{n''} \right) \right\|^2 \right]$$

$$+ 6I\gamma^2 \sum_{n=1}^{N} q_t^n \mathbb{E}_{\cdot|\mathcal{Q}} \left[ \left\| \nabla F_n \left( \sum_{n''=1}^{N} q_t^{n''} \mathbf{y}_{t,i}^{n''} \right) - \sum_{n'=1}^{N} q_t^{n'} \nabla F_{n'} \left( \sum_{n''=1}^{N} q_t^{n''} \mathbf{y}_{t,i}^{n''} \right) \right\|^2 \right]$$

$$+ 6I\gamma^2 \sum_{n=1}^{N} q_t^n \mathbb{E}_{\cdot|\mathcal{Q}} \left[ \left\| \sum_{n'=1}^{N} q_t^{n'} \nabla F_{n'} \left( \sum_{n''=1}^{N} q_t^{n''} \mathbf{y}_{t,i}^{n''} \right) - \sum_{n'=1}^{N} q_t^{n'} \nabla F_{n'} (\mathbf{y}_{t,i}^{n'}) \right\|^2 \right]$$

$$+ \gamma^2 \sigma^2 \left( 1 - \sum_{n=1}^{N} (q_t^n)^2 \right)$$

$$\overset{(c)}{\leq} \left( 1 + \frac{1}{2I-1} \right) \sum_{n=1}^{N} q_t^n \mathbb{E}_{\cdot|\mathcal{Q}} \left[ \left\| \mathbf{y}_{t,i}^n - \sum_{n'=1}^{N} q_t^{n'} \mathbf{y}_{t,i}^{n'} \right\|^2 \right]$$

$$+ 6IL^2\gamma^2 \sum_{n=1}^{N} q_t^n \mathbb{E}_{\cdot|\mathcal{Q}} \left[ \left\| \mathbf{y}_{t,i}^n - \sum_{n''=1}^{N} q_t^{n''} \mathbf{y}_{t,i}^{n''} \right\|^2 \right] + 6I\gamma^2 \tilde{\nu}^2$$

$$+ 6IL^2\gamma^2 \sum_{n=1}^{N} q_t^n \sum_{n'=1}^{N} q_t^{n'} \mathbb{E}_{\cdot|\mathcal{Q}} \left[ \left\| \mathbf{y}_{t,i}^{n'} - \sum_{n''=1}^{N} q_t^{n''} \mathbf{y}_{t,i}^{n''} \right\|^2 \right] + \gamma^2 \sigma^2 \left( 1 - \sum_{n=1}^{N} (q_t^n)^2 \right)$$

$$\overset{(d)}{=} \left( 1 + \frac{1}{2I-1} + 12IL^2\gamma^2 \right) \sum_{n=1}^{N} q_t^n \mathbb{E}_{\cdot|\mathcal{Q}} \left[ \left\| \mathbf{y}_{t,i}^n - \sum_{n'=1}^{N} q_t^{n'} \mathbf{y}_{t,i}^{n'} \right\|^2 \right]$$

$$+ 6I\gamma^2 \tilde{\nu}^2 + \gamma^2 \sigma^2 \left( 1 - \sum_{n=1}^{N} (q_t^n)^2 \right)$$

$$\overset{(e)}{\leq} \left( 1 + \frac{7}{12 \left( I - \frac{1}{2} \right)} \right) \sum_{n=1}^{N} q_t^n \mathbb{E}_{\cdot|\mathcal{Q}} \left[ \left\| \mathbf{y}_{t,i}^n - \sum_{n'=1}^{N} q_t^{n'} \mathbf{y}_{t,i}^{n'} \right\|^2 \right] + 6I\gamma^2 \tilde{\nu}^2 + \gamma^2 \sigma^2 \left( 1 - \sum_{n=1}^{N} (q_t^n)^2 \right).$$

$$\tag{C.6}$$

In the above, $(a)$ is due to $\mathbb{E}_{\cdot|\mathcal{Q}} \left[ \|\mathbf{z}\|^2 \right] = \left\| \mathbb{E}_{\cdot|\mathcal{Q}} [\mathbf{z}] \right\|^2 + \mathbb{E}_{\cdot|\mathcal{Q}} \left[ \|\mathbf{z} - \mathbb{E}_{\cdot|\mathcal{Q}} [\mathbf{z}]\|^2 \right]$; $(b)$ is because the stochastic gradient noise is independent across clients, the variance of the sum of independent random variables is equal to the sum of the variance and $\mathrm{Var}(a\mathbf{z}) = a^2 \mathrm{Var}(\mathbf{z})$ where $\mathrm{Var}(\mathbf{z}) := \mathbb{E}_{\cdot|\mathcal{Q}} \left[ \|\mathbf{z} - \mathbb{E}_{\cdot|\mathcal{Q}} [\mathbf{z}]\|^2 \right]$, and we also use the definition of $\sigma^2$ in Assumption 2 while noting the law of total expectation as well as

$$\sum_{n=1}^{N} q_t^n \left[ \sum_{n' \in \{1,\dots,N\} \setminus \{n\}} (q_t^{n'})^2 + (1 - q_t^n)^2 \right] = \sum_{n=1}^{N} q_t^n \left[ \sum_{n'=1}^{N} (q_t^{n'})^2 - (q_t^n)^2 + (1 - q_t^n)^2 \right]$$

$$= \sum_{n=1}^{N} (q_t^n)^2 + \sum_{n=1}^{N} q_t^n (1 - 2q_t^n) = \sum_{n=1}^{N} (q_t^n)^2 + \sum_{n=1}^{N} q_t^n - 2 \sum_{n=1}^{N} (q_t^n)^2 = 1 - \sum_{n=1}^{N} (q_t^n)^2;$$

$(c)$ uses smoothness in the second term, the definition of $\tilde{\nu}$ in the third term, and Jensen's inequality followed by smoothness in the fourth term, while noting that $\sum_{n=1}^{N} q_t^n = 1$; $(d)$ combines the second and fourth terms in the above line, while (again) noting that $\sum_{n=1}^{N} q_t^n = 1$; $(e)$ is because

$$\frac{1}{2I-1} + 12IL^2\gamma^2 \leq \frac{1}{2I-1} + \frac{1}{12I} \leq \frac{1}{2 \left( I - \frac{1}{2} \right)} + \frac{1}{12 \left( I - \frac{1}{2} \right)} = \frac{7}{12 \left( I - \frac{1}{2} \right)}$$

where the first inequality is due to $\gamma \leq \frac{1}{12LIP} \leq \frac{1}{12LI}$ since $P \geq 1$.

For $i = 0$, we note that

$$\sum_{n=1}^{N} q_t^n \mathbb{E}_{\cdot|\mathcal{Q}} \left[ \left\| \mathbf{y}_{t,0}^n - \sum_{n'=1}^{N} q_t^{n'} \mathbf{y}_{t,0}^{n'} \right\|^2 \right] = 0 \tag{C.7}$$

because $\mathbf{y}_{t,0}^n = \mathbf{x}_t$ for all $n$ as specified in Algorithm 1, and $\sum_{n=1}^N q_t^n = 1$.

By combining (C.6) and (C.7) and unrolling the recursion, we have for any $i$ that

$$\sum_{n=1}^N q_t^n \mathbb{E}_{\cdot|\mathcal{Q}}\left[\left\|\mathbf{y}_{t,i}^n - \sum_{n'=1}^N q_t^{n'} \mathbf{y}_{t,i}^{n'}\right\|^2\right]$$

$$\leq \sum_{i=0}^{I-1}\left(1 + \frac{7}{12\left(I - \frac{1}{2}\right)}\right)^i \left(6I\gamma^2\tilde{\nu}^2 + \gamma^2\sigma^2\left(1 - \sum_{n=1}^N (q_t^n)^2\right)\right)$$

$$\overset{(a)}{=} \left[\left(1 + \frac{7}{12\left(I - \frac{1}{2}\right)}\right)^I - 1\right] \cdot \frac{12\left(I - \frac{1}{2}\right)}{7} \cdot \left(6I\gamma^2\tilde{\nu}^2 + \gamma^2\sigma^2\left(1 - \sum_{n=1}^N (q_t^n)^2\right)\right)$$

$$= \left[\left(1 + \frac{7}{12(I-\frac{1}{2})}\right)^{I-\frac{1}{2}}\left(1 + \frac{7}{12(I-\frac{1}{2})}\right)^{\frac{1}{2}} - 1\right] \cdot \frac{12\left(I - \frac{1}{2}\right)}{7} \cdot \left(6I\gamma^2\tilde{\nu}^2 + \gamma^2\sigma^2\left(1 - \sum_{n=1}^N (q_t^n)^2\right)\right)$$

$$\overset{(b)}{\leq} \left[e^{\frac{7}{12}} \cdot \sqrt{\frac{13}{6}} - 1\right] \cdot \frac{12\left(I - \frac{1}{2}\right)}{7} \cdot \left(6I\gamma^2\tilde{\nu}^2 + \gamma^2\sigma^2\left(1 - \sum_{n=1}^N (q_t^n)^2\right)\right)$$

$$\overset{(c)}{\leq} 3I\gamma^2\left(6I\tilde{\nu}^2 + \sigma^2\left(1 - \sum_{n=1}^N (q_t^n)^2\right)\right)$$

where $(a)$ uses the expression of the sum of geometric progression series; $(b)$ uses $(1 + x)^{\frac{1}{x}} \leq e$ and $I \geq 1$ hence $\left(1 + \frac{7}{12(I-\frac{1}{2})}\right)^{\frac{1}{2}} \leq (1 + \frac{7}{6})^{\frac{1}{2}} = \sqrt{\frac{13}{6}}$; $(c)$ is because $\left[e^{\frac{7}{12}} \cdot \sqrt{\frac{13}{6}} - 1\right]\frac{12}{7} \leq 3$ and $I - \frac{1}{2} \leq I$. $\qquad\square$

**Lemma C.2.** *When $\gamma \leq \frac{1}{12LIP}$, we have*

$$\mathbb{E}_{\cdot|\mathcal{Q}}\left[\left\|\sum_{n=1}^N q_t^n \mathbf{y}_{t,i}^n - \mathbf{x}_{t_0}\right\|^2\right]$$

$$\leq 3\gamma^2 I^2 \tilde{\nu}^2 + 24\gamma^2 I^2 P^2 \tilde{\beta}^2 + 24\gamma^2 I^2 P^2 \mathbb{E}_{\cdot|\mathcal{Q}}\left[\|\nabla f(\mathbf{x}_{t_0})\|^2\right]$$

$$+ 3\gamma^2 IP\sigma^2\left[\rho^2 + \frac{1}{6P}\left(1 - \sum_{n=1}^N (q_t^n)^2\right)\right] \qquad\text{(C.8)}$$

*for $t_0 \leq t \leq t_0 + P - 1$.*

*Proof.* We have

$$\mathbb{E}_{\cdot|\mathcal{Q}}\left[\left\|\sum_{n=1}^N q_t^n \mathbf{y}_{t,i+1}^n - \mathbf{x}_{t_0}\right\|^2\right]$$

$$= \mathbb{E}_{\cdot|\mathcal{Q}}\left[\left\|\sum_{n=1}^N q_t^n\left(\mathbf{y}_{t,i}^n - \gamma\mathbf{g}_n(\mathbf{y}_{t,i}^n)\right) - \mathbf{x}_{t_0}\right\|^2\right]$$

$$= \mathbb{E}_{\cdot|\mathcal{Q}}\left[\left\|\sum_{n=1}^N q_t^n\left(\mathbf{y}_{t,i}^n - \gamma\nabla F_n(\mathbf{y}_{t,i}^n)\right) - \mathbf{x}_{t_0}\right\|^2\right] + \mathbb{E}_{\cdot|\mathcal{Q}}\left[\left\|\sum_{n=1}^N q_t^n\gamma\left(\mathbf{g}_n(\mathbf{y}_{t,i}^n) - \nabla F_n(\mathbf{y}_{t,i}^n)\right)\right\|^2\right]$$

$$\leq \mathbb{E}_{\cdot|\mathcal{Q}}\left[\left\|\sum_{n=1}^N q_t^n\left[\mathbf{y}_{t,i}^n - \gamma\left(\nabla F_n(\mathbf{y}_{t,i}^n) - \nabla F_n\left(\sum_{n'=1}^N q_t^{n'}\mathbf{y}_{t,i}^{n'}\right) + \nabla F_n\left(\sum_{n'=1}^N q_t^{n'}\mathbf{y}_{t,i}^{n'}\right)\right.\right.\right.$$

$$
-\nabla F_n(\mathbf{x}_{t_0}) + \nabla F_n(\mathbf{x}_{t_0}) - \nabla f(\mathbf{x}_{t_0}) + \nabla f(\mathbf{x}_{t_0})\bigg)\bigg] - \mathbf{x}_{t_0}\bigg\|^2\bigg] + \gamma^2\rho^2\sigma^2
$$

$$
\leq \left(1 + \frac{1}{2IP-1}\right)\mathbb{E}_{\cdot|\mathcal{Q}}\left[\bigg\|\sum_{n=1}^N q_t^n \mathbf{y}_{t,i}^n - \mathbf{x}_{t_0}\bigg\|^2\right]
$$

$$
+ 8IP\gamma^2\mathbb{E}_{\cdot|\mathcal{Q}}\left[\bigg\|\sum_{n=1}^N q_t^n\left[\nabla F_n(\mathbf{y}_{t,i}^n) - \nabla F_n\left(\sum_{n'=1}^N q_t^{n'}\mathbf{y}_{t,i}^{n'}\right)\right]\bigg\|^2\right]
$$

$$
+ 8IP\gamma^2\mathbb{E}_{\cdot|\mathcal{Q}}\left[\bigg\|\sum_{n=1}^N q_t^n\left[\nabla F_n\left(\sum_{n'=1}^N q_t^{n'}\mathbf{y}_{t,i}^{n'}\right) - \nabla F_n(\mathbf{x}_{t_0})\right]\bigg\|^2\right]
$$

$$
+ 8IP\gamma^2\mathbb{E}_{\cdot|\mathcal{Q}}\left[\bigg\|\sum_{n=1}^N q_t^n\left[\nabla F_n(\mathbf{x}_{t_0}) - \nabla f(\mathbf{x}_{t_0})\right]\bigg\|^2\right]
$$

$$
+ 8IP\gamma^2\mathbb{E}_{\cdot|\mathcal{Q}}\left[\bigg\|\sum_{n=1}^N q_t^n\nabla f(\mathbf{x}_{t_0})\bigg\|^2\right] + \gamma^2\rho^2\sigma^2
$$

$$
\overset{(a)}{\leq} \left(1 + \frac{1}{2IP-1}\right)\mathbb{E}_{\cdot|\mathcal{Q}}\left[\bigg\|\sum_{n=1}^N q_t^n \mathbf{y}_{t,i}^n - \mathbf{x}_{t_0}\bigg\|^2\right]
$$

$$
+ 8IP\gamma^2\sum_{n=1}^N q_t^n\mathbb{E}_{\cdot|\mathcal{Q}}\left[\bigg\|\nabla F_n(\mathbf{y}_{t,i}^n) - \nabla F_n\left(\sum_{n'=1}^N q_t^{n'}\mathbf{y}_{t,i}^{n'}\right)\bigg\|^2\right]
$$

$$
+ 8IP\gamma^2\sum_{n=1}^N q_t^n\mathbb{E}_{\cdot|\mathcal{Q}}\left[\bigg\|\nabla F_n\left(\sum_{n'=1}^N q_t^{n'}\mathbf{y}_{t,i}^{n'}\right) - \nabla F_n(\mathbf{x}_{t_0})\bigg\|^2\right]
$$

$$
+ 8IP\gamma^2\tilde{\beta}^2 + 8IP\gamma^2\mathbb{E}_{\cdot|\mathcal{Q}}\left[\|\nabla f(\mathbf{x}_{t_0})\|^2\right] + \gamma^2\rho^2\sigma^2
$$

$$
\overset{(b)}{\leq} \left(1 + \frac{1}{2IP-1}\right)\mathbb{E}_{\cdot|\mathcal{Q}}\left[\bigg\|\sum_{n=1}^N q_t^n \mathbf{y}_{t,i}^n - \mathbf{x}_{t_0}\bigg\|^2\right]
$$

$$
+ 8IPL^2\gamma^2\sum_{n=1}^N q_t^n\mathbb{E}_{\cdot|\mathcal{Q}}\left[\bigg\|\mathbf{y}_{t,i}^n - \sum_{n'=1}^N q_t^{n'}\mathbf{y}_{t,i}^{n'}\bigg\|^2\right]
$$

$$
+ 8IPL^2\gamma^2\mathbb{E}_{\cdot|\mathcal{Q}}\left[\bigg\|\sum_{n'=1}^N q_t^{n'}\mathbf{y}_{t,i}^{n'} - \mathbf{x}_{t_0}\bigg\|^2\right]
$$

$$
+ 8IP\gamma^2\tilde{\beta}^2 + 8IP\gamma^2\mathbb{E}_{\cdot|\mathcal{Q}}\left[\|\nabla f(\mathbf{x}_{t_0})\|^2\right] + \gamma^2\rho^2\sigma^2
$$

$$
\overset{(c)}{\leq} \left(1 + \frac{1}{2IP-1} + 8IPL^2\gamma^2\right)\mathbb{E}_{\cdot|\mathcal{Q}}\left[\bigg\|\sum_{n=1}^N q_t^n \mathbf{y}_{t,i}^n - \mathbf{x}_{t_0}\bigg\|^2\right]
$$

$$
+ 24I^2PL^2\gamma^4\left(6I\tilde{\nu}^2 + \sigma^2\left(1 - \sum_{n=1}^N (q_t^n)^2\right)\right)
$$

$$
+ 8IP\gamma^2\tilde{\beta}^2 + 8IP\gamma^2\mathbb{E}_{\cdot|\mathcal{Q}}\left[\|\nabla f(\mathbf{x}_{t_0})\|^2\right] + \gamma^2\rho^2\sigma^2
$$

$$
\overset{(d)}{\leq} \left(1 + \frac{5}{9\left(IP - \frac{1}{2}\right)}\right)\mathbb{E}_{\cdot|\mathcal{Q}}\left[\bigg\|\sum_{n=1}^N q_t^n \mathbf{y}_{t,i}^n - \mathbf{x}_{t_0}\bigg\|^2\right] + 144\gamma^4 I^3 PL^2\tilde{\nu}^2
$$

$$+ 8IP\gamma^2\tilde{\beta}^2 + 8IP\gamma^2\mathbb{E}_{\cdot|\mathcal{Q}}\left[\|\nabla f(\mathbf{x}_{t_0})\|^2\right] + \gamma^2\sigma^2\left[\rho^2 + 24\gamma^2 I^2 PL^2\left(1 - \sum_{n=1}^{N}(q_t^n)^2\right)\right]$$

$$\overset{(e)}{\leq} \left(1 + \frac{5}{9\left(IP - \frac{1}{2}\right)}\right)\mathbb{E}_{\cdot|\mathcal{Q}}\left[\left\|\sum_{n=1}^{N}q_t^n\mathbf{y}_{t,i}^n - \mathbf{x}_{t_0}\right\|^2\right] + \frac{\gamma^2 I\tilde{\nu}^2}{P}$$

$$+ 8IP\gamma^2\tilde{\beta}^2 + 8IP\gamma^2\mathbb{E}_{\cdot|\mathcal{Q}}\left[\|\nabla f(\mathbf{x}_{t_0})\|^2\right] + \gamma^2\sigma^2\left[\rho^2 + \frac{1}{6P}\left(1 - \sum_{n=1}^{N}(q_t^n)^2\right)\right]. \tag{C.9}$$

In the above, the stochastic gradient variance decomposition is the same as the proof of Lemma C.1, while noting that $\sum_{n=1}^{N}(q_t^n)^2 \leq \rho^2$; $(a)$ uses Jensen's inequality, the definition of $\tilde{\beta}^2$, and $\sum_{n=1}^{N}q_t^n = 1$; $(b)$ uses $L$-smoothness; $(c)$ is from Lemma C.1; $(d)$ is due to

$$\frac{1}{2IP - 1} + 8IPL^2\gamma^2 \leq \frac{1}{2IP - 1} + \frac{1}{18IP} \leq \frac{1}{2\left(IP - \frac{1}{2}\right)} + \frac{1}{18\left(IP - \frac{1}{2}\right)} = \frac{5}{9\left(IP - \frac{1}{2}\right)}$$

where the first inequality is due to $\gamma \leq \frac{1}{12LIP}$; $(e)$ uses $\gamma \leq \frac{1}{12LIP}$ in the second and last terms.

We note that, for $t_0 \leq t \leq t_0 + P - 2$, we have

$$\mathbb{E}_{\cdot|\mathcal{Q}}\left[\left\|\sum_{n=1}^{N}q_{t+1}^n\mathbf{y}_{t+1,0}^n - \mathbf{x}_{t_0}\right\|^2\right]$$

$$= \mathbb{E}_{\cdot|\mathcal{Q}}\left[\left\|\sum_{n=1}^{N}q_{t+1}^n\mathbf{x}_{t+1} - \mathbf{x}_{t_0}\right\|^2\right] = \mathbb{E}_{\cdot|\mathcal{Q}}\left[\|\mathbf{x}_{t+1} - \mathbf{x}_{t_0}\|^2\right] = \mathbb{E}_{\cdot|\mathcal{Q}}\left[\left\|\sum_{n=1}^{N}q_t^n\mathbf{y}_{t,I}^n - \mathbf{x}_{t_0}\right\|^2\right] \tag{C.10}$$

because $\mathbf{y}_{t,0}^n = \mathbf{x}_t$ for all $n$ as specified in Algorithm 1, and $\sum_{n=1}^{N}q_t^n = 1$ hence $\mathbf{x}_{t+1} = \mathbf{x}_t + \sum_{n=1}^{N}q_t^n(\mathbf{y}_{t,I}^n - \mathbf{x}_t) = \sum_{n=1}^{N}q_t^n\mathbf{y}_{t,I}^n$ according to Algorithm 1.

Therefore, the same recursion in (C.9) also holds between $t + 1$ and $t$, i.e.,

$$\mathbb{E}_{\cdot|\mathcal{Q}}\left[\left\|\sum_{n=1}^{N}q_{t+1}^n\mathbf{y}_{t+1,1}^n - \mathbf{x}_{t_0}\right\|^2\right]$$

$$\leq \left(1 + \frac{5}{9\left(IP - \frac{1}{2}\right)}\right)\mathbb{E}_{\cdot|\mathcal{Q}}\left[\left\|\sum_{n=1}^{N}q_t^n\mathbf{y}_{t,I}^n - \mathbf{x}_{t_0}\right\|^2\right]$$

$$+ \frac{\gamma^2 I\tilde{\nu}^2}{P} + 8IP\gamma^2\tilde{\beta}^2 + 8IP\gamma^2\mathbb{E}_{\cdot|\mathcal{Q}}\left[\|\nabla f(\mathbf{x}_{t_0})\|^2\right] + \gamma^2\sigma^2\left[\rho^2 + \frac{1}{6P}\left(1 - \sum_{n=1}^{N}(q_t^n)^2\right)\right]. \tag{C.11}$$

When $t = t_0$ and $i = 0$, note that

$$\mathbb{E}_{\cdot|\mathcal{Q}}\left[\left\|\sum_{n=1}^{N}q_{t_0}^n\mathbf{y}_{t_0,0}^n - \mathbf{x}_{t_0}\right\|^2\right] = 0 \tag{C.12}$$

because $\mathbf{y}_{t_0,0}^n = \mathbf{x}_{t_0}$ for all $n$ as specified in Algorithm 1, and $\sum_{n=1}^{N}q_t^n = 1$.

We combine (C.9), (C.11), and (C.12). Since $t_0 \leq t \leq t_0 + P - 1$, unrolling the recursion gives

$$\mathbb{E}_{\cdot|\mathcal{Q}}\left[\left\|\sum_{n=1}^{N}q_t^n\mathbf{y}_{t,i}^n - \mathbf{x}_{t_0}\right\|^2\right]$$

$$\leq \sum_{\kappa=0}^{IP-1} \left(1 + \frac{5}{9\left(IP - \frac{1}{2}\right)}\right)^{\kappa} \left(\frac{\gamma^2 I \tilde{\nu}^2}{P} + 8IP\gamma^2\tilde{\beta}^2 + 8IP\gamma^2 \mathbb{E}_{\cdot|\mathcal{Q}}\left[\|\nabla f(\mathbf{x}_{t_0})\|^2\right]\right.$$

$$\left. + \gamma^2\sigma^2\left[\rho^2 + \frac{1}{6P}\left(1 - \sum_{n=1}^{N}(q_t^n)^2\right)\right]\right)$$

$$= \left[\left(1 + \frac{5}{9\left(IP - \frac{1}{2}\right)}\right)^{IP} - 1\right] \cdot \frac{9\left(IP - \frac{1}{2}\right)}{5} \cdot \left(\frac{\gamma^2 I \tilde{\nu}^2}{P} + 8IP\gamma^2\tilde{\beta}^2 + 8IP\gamma^2 \mathbb{E}_{\cdot|\mathcal{Q}}\left[\|\nabla f(\mathbf{x}_{t_0})\|^2\right]\right.$$

$$\left. + \gamma^2\sigma^2\left[\rho^2 + \frac{1}{6P}\left(1 - \sum_{n=1}^{N}(q_t^n)^2\right)\right]\right)$$

$$= \left[\left(1 + \frac{5}{9\left(IP - \frac{1}{2}\right)}\right)^{IP - \frac{1}{2}}\left(1 + \frac{5}{9\left(IP - \frac{1}{2}\right)}\right)^{\frac{1}{2}} - 1\right] \cdot \frac{9\left(IP - \frac{1}{2}\right)}{5} \cdot \left(\frac{\gamma^2 I \tilde{\nu}^2}{P} + 8IP\gamma^2\tilde{\beta}^2\right.$$

$$\left. + 8IP\gamma^2 \mathbb{E}_{\cdot|\mathcal{Q}}\left[\|\nabla f(\mathbf{x}_{t_0})\|^2\right] + \gamma^2\sigma^2\left[\rho^2 + \frac{1}{6P}\left(1 - \sum_{n=1}^{N}(q_t^n)^2\right)\right]\right)$$

$$\leq \left[e^{\frac{5}{9}} \cdot \sqrt{\frac{19}{9}} - 1\right] \cdot \frac{9\left(IP - \frac{1}{2}\right)}{5} \cdot \left(\frac{\gamma^2 I \tilde{\nu}^2}{P} + 8IP\gamma^2\tilde{\beta}^2 + 8IP\gamma^2 \mathbb{E}_{\cdot|\mathcal{Q}}\left[\|\nabla f(\mathbf{x}_{t_0})\|^2\right]\right.$$

$$\left. + \gamma^2\sigma^2\left[\rho^2 + \frac{1}{6P}\left(1 - \sum_{n=1}^{N}(q_t^n)^2\right)\right]\right)$$

$$\leq 3IP\left(\frac{\gamma^2 I \tilde{\nu}^2}{P} + 8IP\gamma^2\tilde{\beta}^2 + 8IP\gamma^2 \mathbb{E}_{\cdot|\mathcal{Q}}\left[\|\nabla f(\mathbf{x}_{t_0})\|^2\right] + \gamma^2\sigma^2\left[\rho^2 + \frac{1}{6P}\left(1 - \sum_{n=1}^{N}(q_t^n)^2\right)\right]\right)$$

$$\leq 3\gamma^2 I^2\tilde{\nu}^2 + 24\gamma^2 I^2 P^2\tilde{\beta}^2 + 24\gamma^2 I^2 P^2 \mathbb{E}_{\cdot|\mathcal{Q}}\left[\|\nabla f(\mathbf{x}_{t_0})\|^2\right] + 3\gamma^2 IP\sigma^2\left[\rho^2 + \frac{1}{6P}\left(1 - \sum_{n=1}^{N}(q_t^n)^2\right)\right]$$

where the rationale behind the steps is similar to the proof of Lemma C.1. $\qquad\square$

### C.3 Proof of Theorem 3.1

Consider $t_0 = kP$ for $k = 0, 1, 2, \dots$.

Due to smoothness,

$$\mathbb{E}_{\cdot|\mathcal{Q},t_0}\left[f(\mathbf{x}_{t_0+P})\right]$$

$$\leq f(\mathbf{x}_{t_0}) + \mathbb{E}_{\cdot|\mathcal{Q},t_0}\left[\langle\nabla f(\mathbf{x}_{t_0}), \mathbf{x}_{t_0+P} - \mathbf{x}_{t_0}\rangle\right] + \frac{L}{2}\mathbb{E}_{\cdot|\mathcal{Q},t_0}\left[\|\mathbf{x}_{t_0+P} - \mathbf{x}_{t_0}\|^2\right]$$

$$\leq f(\mathbf{x}_{t_0}) - \gamma\eta\left\langle\nabla f(\mathbf{x}_{t_0}), \mathbb{E}_{\cdot|\mathcal{Q},t_0}\left[\sum_{t=t_0}^{t_0+P-1}\sum_{n=1}^{N}q_t^n\sum_{i=0}^{I-1}\mathbf{g}_n(\mathbf{y}_{t,i}^n)\right]\right\rangle$$

$$+ \frac{\gamma^2\eta^2 L}{2}\mathbb{E}_{\cdot|\mathcal{Q},t_0}\left[\left\|\sum_{t=t_0}^{t_0+P-1}\sum_{n=1}^{N}q_t^n\sum_{i=0}^{I-1}\mathbf{g}_n(\mathbf{y}_{t,i}^n)\right\|^2\right]$$

$$\overset{(a)}{=} f(\mathbf{x}_{t_0}) - \gamma\eta\left\langle\nabla f(\mathbf{x}_{t_0}), \mathbb{E}_{\cdot|\mathcal{Q},t_0}\left[\sum_{t=t_0}^{t_0+P-1}\sum_{n=1}^{N}q_t^n\sum_{i=0}^{I-1}\mathbb{E}\left[\mathbf{g}_n(\mathbf{y}_{t,i}^n)\big|\mathcal{Q}, \mathbf{y}_{t,i}^n, \mathbf{x}_{t_0}\right]\right]\right\rangle$$

$$+ \frac{\gamma^2\eta^2 L}{2}\mathbb{E}_{\cdot|\mathcal{Q},t_0}\left[\left\|\sum_{t=t_0}^{t_0+P-1}\sum_{n=1}^{N}q_t^n\sum_{i=0}^{I-1}\mathbf{g}_n(\mathbf{y}_{t,i}^n)\right\|^2\right]$$

$$\overset{(b)}{=} f(\mathbf{x}_{t_0}) - \gamma\eta\left\langle\nabla f(\mathbf{x}_{t_0}), \mathbb{E}_{\cdot|\mathcal{Q},t_0}\left[\sum_{t=t_0}^{t_0+P-1}\sum_{n=1}^{N}q_t^n\sum_{i=0}^{I-1}\nabla F_n(\mathbf{y}_{t,i}^n)\right]\right\rangle$$

$$+ \frac{\gamma^2 \eta^2 L}{2} \mathbb{E}_{\cdot|\mathcal{Q},t_0} \left[ \left\| \sum_{t=t_0}^{t_0+P-1} \sum_{n=1}^{N} q_t^n \sum_{i=0}^{I-1} \mathbf{g}_n(\mathbf{y}_{t,i}^n) \right\|^2 \right]$$

where $\mathbb{E}_{\cdot|\mathcal{Q},t_0}[\mathbf{z}]$ is a short-hand notation for $\mathbb{E}[\mathbf{z}|\mathcal{Q}, \mathbf{x}_{t_0}]$; $(a)$ is due to the law of total expectation, where the expectation is taken over $\mathbf{y}_{t,i}^n$; $(b)$ is due to the unbiasedness and independence of stochastic gradients. In other parts of the proof, we may use the law of total expectation in a similar way without explanation.

Taking expectation on both sides over $\mathbf{x}_{t_0}$, we obtain

$$\mathbb{E}_{\cdot|\mathcal{Q}}[f(\mathbf{x}_{t_0+P})] \le \mathbb{E}_{\cdot|\mathcal{Q}}[f(\mathbf{x}_{t_0})] - \mathbb{E}_{\cdot|\mathcal{Q}} \left[ \gamma\eta \left\langle \nabla f(\mathbf{x}_{t_0}), \sum_{t=t_0}^{t_0+P-1} \sum_{n=1}^{N} q_t^n \sum_{i=0}^{I-1} \nabla F_n(\mathbf{y}_{t,i}^n) \right\rangle \right]$$
$$+ \frac{\gamma^2 \eta^2 L}{2} \mathbb{E}_{\cdot|\mathcal{Q}} \left[ \left\| \sum_{t=t_0}^{t_0+P-1} \sum_{n=1}^{N} q_t^n \sum_{i=0}^{I-1} \mathbf{g}_n(\mathbf{y}_{t,i}^n) \right\|^2 \right]. \tag{C.13}$$

Consider the second term in (C.13),

$$- \gamma\eta \left\langle \nabla f(\mathbf{x}_{t_0}), \sum_{t=t_0}^{t_0+P-1} \sum_{n=1}^{N} q_t^n \sum_{i=0}^{I-1} \nabla F_n(\mathbf{y}_{t,i}^n) \right\rangle$$

$$= - \frac{\gamma\eta}{IP} \left\langle IP \nabla f(\mathbf{x}_{t_0}), \sum_{t=t_0}^{t_0+P-1} \sum_{n=1}^{N} q_t^n \sum_{i=0}^{I-1} \nabla F_n(\mathbf{y}_{t,i}^n) \right\rangle$$

$$\overset{(a)}{=} \frac{\gamma\eta}{2IP} \left\| \sum_{t=t_0}^{t_0+P-1} \sum_{n=1}^{N} q_t^n \sum_{i=0}^{I-1} \nabla F_n(\mathbf{y}_{t,i}^n) - IP \nabla f(\mathbf{x}_{t_0}) \right\|^2$$
$$- \frac{\gamma\eta IP}{2} \|\nabla f(\mathbf{x}_{t_0})\|^2 - \frac{\gamma\eta}{2IP} \left\| \sum_{t=t_0}^{t_0+P-1} \sum_{n=1}^{N} q_t^n \sum_{i=0}^{I-1} \nabla F_n(\mathbf{y}_{t,i}^n) \right\|^2$$

$$= \frac{\gamma\eta}{2IP} \left\| \sum_{t=t_0}^{t_0+P-1} \sum_{n=1}^{N} q_t^n \sum_{i=0}^{I-1} \left[ \nabla F_n(\mathbf{y}_{t,i}^n) - \nabla f(\mathbf{x}_{t_0}) \right] \right\|^2$$
$$- \frac{\gamma\eta IP}{2} \|\nabla f(\mathbf{x}_{t_0})\|^2 - \frac{\gamma\eta}{2IP} \left\| \sum_{t=t_0}^{t_0+P-1} \sum_{n=1}^{N} q_t^n \sum_{i=0}^{I-1} \nabla F_n(\mathbf{y}_{t,i}^n) \right\|^2$$

$$= \frac{\gamma\eta}{2IP} \left\| \sum_{t=t_0}^{t_0+P-1} \sum_{n=1}^{N} q_t^n \sum_{i=0}^{I-1} \left[ \nabla F_n(\mathbf{y}_{t,i}^n) - \nabla F_n \left( \sum_{n'=1}^{N} q_t^{n'} \mathbf{y}_{t,i}^{n'} \right) + \nabla F_n \left( \sum_{n'=1}^{N} q_t^{n'} \mathbf{y}_{t,i}^{n'} \right) \right. \right.$$
$$\left. \left. - \nabla F_n(\mathbf{x}_{t_0}) + \nabla F_n(\mathbf{x}_{t_0}) - \nabla f(\mathbf{x}_{t_0}) \right] \right\|^2$$

$$- \frac{\gamma\eta IP}{2} \|\nabla f(\mathbf{x}_{t_0})\|^2 - \frac{\gamma\eta}{2IP} \left\| \sum_{t=t_0}^{t_0+P-1} \sum_{n=1}^{N} q_t^n \sum_{i=0}^{I-1} \nabla F_n(\mathbf{y}_{t,i}^n) \right\|^2$$

$$\overset{(b)}{\le} \frac{3\gamma\eta}{2IP} \left\| \sum_{t=t_0}^{t_0+P-1} \sum_{n=1}^{N} q_t^n \sum_{i=0}^{I-1} \left[ \nabla F_n(\mathbf{y}_{t,i}^n) - \nabla F_n \left( \sum_{n'=1}^{N} q_t^{n'} \mathbf{y}_{t,i}^{n'} \right) \right] \right\|^2$$
$$+ \frac{3\gamma\eta}{2IP} \left\| \sum_{t=t_0}^{t_0+P-1} \sum_{n=1}^{N} q_t^n \sum_{i=0}^{I-1} \left[ \nabla F_n \left( \sum_{n'=1}^{N} q_t^{n'} \mathbf{y}_{t,i}^{n'} \right) - \nabla F_n(\mathbf{x}_{t_0}) \right] \right\|^2$$
$$+ \frac{3\gamma\eta}{2IP} \left\| \sum_{t=t_0}^{t_0+P-1} \sum_{n=1}^{N} q_t^n \sum_{i=0}^{I-1} \left[ \nabla F_n(\mathbf{x}_{t_0}) - \nabla f(\mathbf{x}_{t_0}) \right] \right\|^2$$

$$-\frac{\gamma\eta IP}{2}\|\nabla f(\mathbf{x}_{t_0})\|^2-\frac{\gamma\eta}{2IP}\left\|\sum_{t=t_0}^{t_0+P-1}\sum_{n=1}^{N}q_t^n\sum_{i=0}^{I-1}\nabla F_n(\mathbf{y}_{t,i}^n)\right\|^2$$

$$\overset{(c)}{\le}\frac{3\gamma\eta}{2}\sum_{t=t_0}^{t_0+P-1}\sum_{n=1}^{N}\sum_{i=0}^{I-1}q_t^n\left\|\nabla F_n(\mathbf{y}_{t,i}^n)-\nabla F_n\left(\sum_{n'=1}^{N}q_t^{n'}\mathbf{y}_{t,i}^{n'}\right)\right\|^2$$

$$+\frac{3\gamma\eta}{2}\sum_{t=t_0}^{t_0+P-1}\sum_{n=1}^{N}\sum_{i=0}^{I-1}q_t^n\left\|\nabla F_n\left(\sum_{n'=1}^{N}q_t^{n'}\mathbf{y}_{t,i}^{n'}\right)-\nabla F_n(\mathbf{x}_{t_0})\right\|^2$$

$$+\frac{3\gamma\eta IP\tilde{\delta}^2(P)}{2}-\frac{\gamma\eta IP}{2}\|\nabla f(\mathbf{x}_{t_0})\|^2-\frac{\gamma\eta}{2IP}\left\|\sum_{t=t_0}^{t_0+P-1}\sum_{n=1}^{N}q_t^n\sum_{i=0}^{I-1}\nabla F_n(\mathbf{y}_{t,i}^n)\right\|^2$$

$$\le\frac{3\gamma\eta L^2}{2}\sum_{t=t_0}^{t_0+P-1}\sum_{n=1}^{N}\sum_{i=0}^{I-1}q_t^n\left\|\mathbf{y}_{t,i}^n-\sum_{n'=1}^{N}q_t^{n'}\mathbf{y}_{t,i}^{n'}\right\|^2$$

$$+\frac{3\gamma\eta L^2}{2}\sum_{t=t_0}^{t_0+P-1}\sum_{i=0}^{I-1}\left\|\sum_{n'=1}^{N}q_t^{n'}\mathbf{y}_{t,i}^{n'}-\mathbf{x}_{t_0}\right\|^2$$

$$+\frac{3\gamma\eta IP\tilde{\delta}^2(P)}{2}-\frac{\gamma\eta IP}{2}\|\nabla f(\mathbf{x}_{t_0})\|^2-\frac{\gamma\eta}{2IP}\left\|\sum_{t=t_0}^{t_0+P-1}\sum_{n=1}^{N}q_t^n\sum_{i=0}^{I-1}\nabla F_n(\mathbf{y}_{t,i}^n)\right\|^2$$

where $(a)$ uses $-\langle\mathbf{a},\mathbf{b}\rangle=\frac{1}{2}(\|\mathbf{a}-\mathbf{b}\|^2-\|\mathbf{a}\|^2-\|\mathbf{b}\|^2)$; $(b)$ uses Jensen's inequality in the first term; $(c)$ uses Jensen's inequality in the first two terms and the definition of $\tilde{\delta}^2(P)$ in the third term. Hence,

$$-\mathbb{E}_{\cdot|\mathcal{Q}}\left[\gamma\eta\left\langle\nabla f(\mathbf{x}_{t_0}),\sum_{t=t_0}^{t_0+P-1}\sum_{n=1}^{N}q_t^n\sum_{i=0}^{I-1}\nabla F_n(\mathbf{y}_{t,i}^n)\right\rangle\right]$$

$$\le\frac{3\gamma\eta L^2}{2}\sum_{t=t_0}^{t_0+P-1}\sum_{n=1}^{N}\sum_{i=0}^{I-1}q_t^n\mathbb{E}_{\cdot|\mathcal{Q}}\left[\left\|\mathbf{y}_{t,i}^n-\sum_{n'=1}^{N}q_t^{n'}\mathbf{y}_{t,i}^{n'}\right\|^2\right]$$

$$+\frac{3\gamma\eta L^2}{2}\sum_{t=t_0}^{t_0+P-1}\sum_{i=0}^{I-1}\mathbb{E}_{\cdot|\mathcal{Q}}\left[\left\|\sum_{n'=1}^{N}q_t^{n'}\mathbf{y}_{t,i}^{n'}-\mathbf{x}_{t_0}\right\|^2\right]$$

$$+\frac{3\gamma\eta IP\tilde{\delta}^2(P)}{2}-\frac{\gamma\eta IP}{2}\mathbb{E}_{\cdot|\mathcal{Q}}\left[\|\nabla f(\mathbf{x}_{t_0})\|^2\right]-\frac{\gamma\eta}{2IP}\mathbb{E}_{\cdot|\mathcal{Q}}\left[\left\|\sum_{t=t_0}^{t_0+P-1}\sum_{n=1}^{N}q_t^n\sum_{i=0}^{I-1}\nabla F_n(\mathbf{y}_{t,i}^n)\right\|^2\right]$$

$$\overset{(a)}{\le}\frac{3\gamma\eta L^2}{2}\sum_{t=t_0}^{t_0+P-1}\sum_{i=0}^{I-1}3I\gamma^2\left(6I\tilde{\nu}^2+\sigma^2\left(1-\sum_{n=1}^{N}(q_t^n)^2\right)\right)$$

$$+\frac{3\gamma\eta L^2}{2}\sum_{t=t_0}^{t_0+P-1}\sum_{i=0}^{I-1}\left(3\gamma^2 I^2\tilde{\nu}^2+24\gamma^2 I^2 P^2\tilde{\beta}^2+24\gamma^2 I^2 P^2\mathbb{E}_{\cdot|\mathcal{Q}}\left[\|\nabla f(\mathbf{x}_{t_0})\|^2\right]\right.$$

$$\left.+3\gamma^2 IP\sigma^2\left[\rho^2+\frac{1}{6P}\left(1-\sum_{n=1}^{N}(q_t^n)^2\right)\right]\right)$$

$$+\frac{3\gamma\eta IP\tilde{\delta}^2(P)}{2}-\frac{\gamma\eta IP}{2}\mathbb{E}_{\cdot|\mathcal{Q}}\left[\|\nabla f(\mathbf{x}_{t_0})\|^2\right]-\frac{\gamma\eta}{2IP}\mathbb{E}_{\cdot|\mathcal{Q}}\left[\left\|\sum_{t=t_0}^{t_0+P-1}\sum_{n=1}^{N}q_t^n\sum_{i=0}^{I-1}\nabla F_n(\mathbf{y}_{t,i}^n)\right\|^2\right]$$

$$=\frac{9\gamma^3\eta L^2 I^2 P}{2}\left(6I\tilde{\nu}^2+\sigma^2\left(1-\sum_{n=1}^{N}(q_t^n)^2\right)\right)$$

$$+ \frac{3\gamma\eta L^2 IP}{2} \left( 3\gamma^2 I^2 \tilde{\nu}^2 + 24\gamma^2 I^2 P^2 \tilde{\beta}^2 + 24\gamma^2 I^2 P^2 \mathbb{E}_{\cdot|\mathcal{Q}} \left[ \|\nabla f(\mathbf{x}_{t_0})\|^2 \right] \right.$$

$$\left. + 3\gamma^2 IP\sigma^2 \left[ \rho^2 + \frac{1}{6P} \left( 1 - \sum_{n=1}^{N} (q_t^n)^2 \right) \right] \right)$$

$$+ \frac{3\gamma\eta IP\tilde{\delta}^2(P)}{2} - \frac{\gamma\eta IP}{2} \mathbb{E}_{\cdot|\mathcal{Q}} \left[ \|\nabla f(\mathbf{x}_{t_0})\|^2 \right] - \frac{\gamma\eta}{2IP} \mathbb{E}_{\cdot|\mathcal{Q}} \left[ \left\| \sum_{t=t_0}^{t_0+P-1} \sum_{n=1}^{N} q_t^n \sum_{i=0}^{I-1} \nabla F_n(\mathbf{y}_{t,i}^n) \right\|^2 \right] \tag{C.14}$$

where $(a)$ uses Lemmas C.1 and C.2.

Note also that

$$\mathbb{E}_{\cdot|\mathcal{Q}} \left[ \left\| \sum_{t=t_0}^{t_0+P-1} \sum_{n=1}^{N} q_t^n \sum_{i=0}^{I-1} \mathbf{g}_n(\mathbf{y}_{t,i}^n) \right\|^2 \right]$$

$$\overset{(a)}{=} \mathbb{E}_{\cdot|\mathcal{Q}} \left[ \left\| \sum_{t=t_0}^{t_0+P-1} \sum_{n=1}^{N} q_t^n \sum_{i=0}^{I-1} \nabla F_n(\mathbf{y}_{t,i}^n) \right\|^2 \right]$$

$$+ \mathbb{E}_{\cdot|\mathcal{Q}} \left[ \left\| \sum_{t=t_0}^{t_0+P-1} \sum_{n=1}^{N} q_t^n \sum_{i=0}^{I-1} \left[ \mathbf{g}_n(\mathbf{y}_{t,i}^n) - \nabla F_n(\mathbf{y}_{t,i}^n) \right] \right\|^2 \right]$$

$$\overset{(b)}{=} \mathbb{E}_{\cdot|\mathcal{Q}} \left[ \left\| \sum_{t=t_0}^{t_0+P-1} \sum_{n=1}^{N} q_t^n \sum_{i=0}^{I-1} \nabla F_n(\mathbf{y}_{t,i}^n) \right\|^2 \right]$$

$$+ \sum_{t=t_0}^{t_0+P-1} \sum_{n=1}^{N} (q_t^n)^2 \sum_{i=0}^{I-1} \mathbb{E}_{\cdot|\mathcal{Q}} \left[ \|\mathbf{g}_n(\mathbf{y}_{t,i}^n) - \nabla F_n(\mathbf{y}_{t,i}^n)\|^2 \right]$$

$$\overset{(c)}{\leq} \mathbb{E}_{\cdot|\mathcal{Q}} \left[ \left\| \sum_{t=t_0}^{t_0+P-1} \sum_{n=1}^{N} q_t^n \sum_{i=0}^{I-1} \nabla F_n(\mathbf{y}_{t,i}^n) \right\|^2 \right] + IP\rho^2\sigma^2 \tag{C.15}$$

where $(a)$ uses $\mathbb{E}_{\cdot|\mathcal{Q}} \left[ \|\mathbf{z}\|^2 \right] = \left\| \mathbb{E}_{\cdot|\mathcal{Q}} [\mathbf{z}] \right\|^2 + \mathbb{E}_{\cdot|\mathcal{Q}} \left[ \|\mathbf{z} - \mathbb{E}_{\cdot|\mathcal{Q}} [\mathbf{z}]\|^2 \right]$; $(b)$ is because the variance of the sum of independent random variables is equal to the sum of the variance and $\text{Var}(a\mathbf{z}) = a^2 \text{Var}(\mathbf{z})$ where $\text{Var}(\mathbf{z}) := \mathbb{E}_{\cdot|\mathcal{Q}} \left[ \|\mathbf{z} - \mathbb{E}_{\cdot|\mathcal{Q}} [\mathbf{z}]\|^2 \right]$; $(c)$ uses the definitions of $\sigma^2$ and $\rho^2$.

Plugging (C.14) and (C.15) back into (C.13), we get

$$\mathbb{E}_{\cdot|\mathcal{Q}} \left[ f(\mathbf{x}_{t_0+P}) \right]$$

$$\leq \mathbb{E}_{\cdot|\mathcal{Q}} \left[ f(\mathbf{x}_{t_0}) \right] - \mathbb{E}_{\cdot|\mathcal{Q}} \left[ \gamma\eta \left\langle \nabla f(\mathbf{x}_{t_0}), \sum_{t=t_0}^{t_0+P-1} \sum_{n=1}^{N} q_t^n \sum_{i=0}^{I-1} \nabla F_n(\mathbf{y}_{t,i}^n) \right\rangle \right]$$

$$+ \frac{\gamma^2\eta^2 L}{2} \mathbb{E}_{\cdot|\mathcal{Q}} \left[ \left\| \sum_{t=t_0}^{t_0+P-1} \sum_{n=1}^{N} q_t^n \sum_{i=0}^{I-1} \mathbf{g}_n(\mathbf{y}_{t,i}^n) \right\|^2 \right]$$

$$\leq \mathbb{E}_{\cdot|\mathcal{Q}} \left[ f(\mathbf{x}_{t_0}) \right] + \frac{9\gamma^3\eta L^2 I^2 P}{2} \left( 6I\tilde{\nu}^2 + \sigma^2 \left( 1 - \sum_{n=1}^{N} (q_t^n)^2 \right) \right)$$

$$+ \frac{3\gamma\eta L^2 IP}{2} \left( 3\gamma^2 I^2 \tilde{\nu}^2 + 24\gamma^2 I^2 P^2 \tilde{\beta}^2 + 24\gamma^2 I^2 P^2 \mathbb{E}_{\cdot|\mathcal{Q}} \left[ \|\nabla f(\mathbf{x}_{t_0})\|^2 \right] \right.$$

$$\left. + 3\gamma^2 IP\sigma^2 \left[ \rho^2 + \frac{1}{6P} \left( 1 - \sum_{n=1}^{N} (q_t^n)^2 \right) \right] \right)$$

$$+ \frac{3\gamma\eta IP\tilde{\delta}^2(P)}{2} - \frac{\gamma\eta IP}{2}\mathbb{E}_{\cdot|\mathcal{Q}}\left[\|\nabla f(\mathbf{x}_{t_0})\|^2\right] - \frac{\gamma\eta}{2IP}\mathbb{E}_{\cdot|\mathcal{Q}}\left[\left\|\sum_{t=t_0}^{t_0+P-1}\sum_{n=1}^{N}q_t^n\sum_{i=0}^{I-1}\nabla F_n(\mathbf{y}_{t,i}^n)\right\|^2\right]$$

$$+ \frac{\gamma^2\eta^2 L}{2}\mathbb{E}_{\cdot|\mathcal{Q}}\left[\left\|\sum_{t=t_0}^{t_0+P-1}\sum_{n=1}^{N}q_t^n\sum_{i=0}^{I-1}\nabla F_n(\mathbf{y}_{t,i}^n)\right\|^2\right] + \frac{\gamma^2\eta^2 LIP\rho^2\sigma^2}{2}$$

$$= \mathbb{E}_{\cdot|\mathcal{Q}}\left[f(\mathbf{x}_{t_0})\right] + \frac{9\gamma^3\eta L^2 I^2 P}{2}\left(6I\tilde{\nu}^2 + \sigma^2\left(1 - \sum_{n=1}^{N}(q_t^n)^2\right)\right)$$

$$+ \frac{3\gamma\eta L^2 IP}{2}\left(3\gamma^2 I^2\tilde{\nu}^2 + 24\gamma^2 I^2 P^2\tilde{\beta}^2 + 3\gamma^2 IP\sigma^2\left[\rho^2 + \frac{1}{6P}\left(1 - \sum_{n=1}^{N}(q_t^n)^2\right)\right]\right)$$

$$+ \frac{3\gamma\eta IP\tilde{\delta}^2(P)}{2} - \frac{\gamma\eta IP}{2}\left(1 - 72\gamma^2 L^2 I^2 P^2\right)\mathbb{E}_{\cdot|\mathcal{Q}}\left[\|\nabla f(\mathbf{x}_{t_0})\|^2\right]$$

$$- \left(\frac{\gamma\eta}{2IP} - \frac{\gamma^2\eta^2 L}{2}\right)\mathbb{E}_{\cdot|\mathcal{Q}}\left[\left\|\sum_{t=t_0}^{t_0+P-1}\sum_{n=1}^{N}q_t^n\sum_{i=0}^{I-1}\nabla F_n(\mathbf{y}_{t,i}^n)\right\|^2\right]$$

$$+ \frac{\gamma^2\eta^2 LIP\rho^2\sigma^2}{2}$$

$$\leq \mathbb{E}_{\cdot|\mathcal{Q}}\left[f(\mathbf{x}_{t_0})\right] + \frac{9\gamma^3\eta L^2 I^2 P}{2}\left(6I\tilde{\nu}^2 + \sigma^2\left(1 - \sum_{n=1}^{N}(q_t^n)^2\right)\right)$$

$$+ \frac{3\gamma\eta L^2 IP}{2}\left(3\gamma^2 I^2\tilde{\nu}^2 + 24\gamma^2 I^2 P^2\tilde{\beta}^2 + 3\gamma^2 IP\sigma^2\left[\rho^2 + \frac{1}{6P}\left(1 - \sum_{n=1}^{N}(q_t^n)^2\right)\right]\right)$$

$$+ \frac{3\gamma\eta IP\tilde{\delta}^2(P)}{2} - \frac{\gamma\eta IP}{4}\mathbb{E}_{\cdot|\mathcal{Q}}\left[\|\nabla f(\mathbf{x}_{t_0})\|^2\right]$$

$$+ \frac{\gamma^2\eta^2 LIP\rho^2\sigma^2}{2}$$

where the last inequality is because $\gamma \leq \frac{1}{12LIP}$ thus $-(1 - 72\gamma^2 L^2 I^2 P^2) \leq -\frac{1}{2}$, and $\gamma\eta \leq \frac{1}{LIP}$ thus $\frac{1}{IP} - \gamma\eta L \geq 0$.

Rearranging, we have

$$\frac{\gamma\eta IP}{4}\mathbb{E}_{\cdot|\mathcal{Q}}\left[\|\nabla f(\mathbf{x}_{t_0})\|^2\right]$$

$$\leq \mathbb{E}_{\cdot|\mathcal{Q}}\left[f(\mathbf{x}_{t_0})\right] - \mathbb{E}_{\cdot|\mathcal{Q}}\left[f(\mathbf{x}_{t_0+P})\right] + \frac{9\gamma^3\eta L^2 I^2 P}{2}\left(6I\tilde{\nu}^2 + \sigma^2\left(1 - \sum_{n=1}^{N}(q_t^n)^2\right)\right)$$

$$+ \frac{3\gamma\eta L^2 IP}{2}\left(3\gamma^2 I^2\tilde{\nu}^2 + 24\gamma^2 I^2 P^2\tilde{\beta}^2 + 3\gamma^2 IP\sigma^2\left[\rho^2 + \frac{1}{6P}\left(1 - \sum_{n=1}^{N}(q_t^n)^2\right)\right]\right)$$

$$+ \frac{3\gamma\eta IP\tilde{\delta}^2(P)}{2} + \frac{\gamma^2\eta^2 LIP\rho^2\sigma^2}{2}.$$

Hence,

$$\mathbb{E}_{\cdot|\mathcal{Q}}\left[\|\nabla f(\mathbf{x}_{t_0})\|^2\right]$$

$$\leq \frac{4\left(\mathbb{E}_{\cdot|\mathcal{Q}}\left[f(\mathbf{x}_{t_0})\right] - \mathbb{E}_{\cdot|\mathcal{Q}}\left[f(\mathbf{x}_{t_0+P})\right]\right)}{\gamma\eta IP} + 18\gamma^2 L^2 I\left(6I\tilde{\nu}^2 + \sigma^2\left(1 - \sum_{n=1}^{N}(q_t^n)^2\right)\right)$$

$$+ 6L^2\left(3\gamma^2 I^2\tilde{\nu}^2 + 24\gamma^2 I^2 P^2\tilde{\beta}^2 + 3\gamma^2 IP\sigma^2\left[\rho^2 + \frac{1}{6P}\left(1 - \sum_{n=1}^{N}(q_t^n)^2\right)\right]\right)$$

$$+ 6\tilde{\delta}^2(P) + 2\gamma\eta L\rho^2\sigma^2$$

$$= \frac{4\left(\mathbb{E}_{\cdot|\mathcal{Q}}\left[f(\mathbf{x}_{t_0})\right] - \mathbb{E}_{\cdot|\mathcal{Q}}\left[f(\mathbf{x}_{t_0+P})\right]\right)}{\gamma\eta IP} + 126\gamma^2 L^2 I^2\tilde{\nu}^2 + 144\gamma^2 L^2 I^2 P^2\tilde{\beta}^2 + 6\tilde{\delta}^2(P)$$

$$+ 18\gamma^2 L^2 I\sigma^2\left(1 - \sum_{n=1}^{N}(q_t^n)^2\right) + 18\gamma^2 L^2 IP\sigma^2\left[\rho^2 + \frac{1}{6P}\left(1 - \sum_{n=1}^{N}(q_t^n)^2\right)\right] + 2\gamma\eta L\rho^2\sigma^2$$

$$= \frac{4\left(\mathbb{E}_{\cdot|\mathcal{Q}}\left[f(\mathbf{x}_{t_0})\right] - \mathbb{E}_{\cdot|\mathcal{Q}}\left[f(\mathbf{x}_{t_0+P})\right]\right)}{\gamma\eta IP} + 126\gamma^2 L^2 I^2\tilde{\nu}^2 + 144\gamma^2 L^2 I^2 P^2\tilde{\beta}^2 + 6\tilde{\delta}^2(P)$$

$$+ 21\gamma^2 L^2 I\sigma^2\left(1 - \sum_{n=1}^{N}(q_t^n)^2\right) + 18\gamma^2 L^2 IP\sigma^2\rho^2 + 2\gamma\eta L\rho^2\sigma^2$$

$$\leq \frac{4\left(\mathbb{E}_{\cdot|\mathcal{Q}}\left[f(\mathbf{x}_{t_0})\right] - \mathbb{E}_{\cdot|\mathcal{Q}}\left[f(\mathbf{x}_{t_0+P})\right]\right)}{\gamma\eta IP} + 126\gamma^2 L^2 I^2\tilde{\nu}^2 + 144\gamma^2 L^2 I^2 P^2\tilde{\beta}^2 + 6\tilde{\delta}^2(P)$$

$$+ \left(21\gamma^2 L^2 I + 18\gamma^2 L^2 IP\rho^2 + 2\gamma\eta L\rho^2\right)\sigma^2. \tag{C.16}$$

We now have

$$\min_t \mathbb{E}_{\cdot|\mathcal{Q}}\left[\|\nabla f(\mathbf{x}_t)\|^2\right]$$

$$\leq \min_{t_0 \in \left\{0, P, 2P, \ldots, \left(\lfloor\frac{T}{P}\rfloor - 1\right)P\right\}} \mathbb{E}_{\cdot|\mathcal{Q}}\left[\|\nabla f(\mathbf{x}_{t_0})\|^2\right]$$

$$\leq \frac{1}{\lfloor T/P\rfloor} \cdot \sum_{t_0 = 0, P, 2P, \ldots, \left(\lfloor\frac{T}{P}\rfloor - 1\right)P} \mathbb{E}_{\cdot|\mathcal{Q}}\left[\|\nabla f(\mathbf{x}_{t_0})\|^2\right]$$

$$\overset{(a)}{\leq} \frac{8\left(f(\mathbf{x}_0) - f^*\right)}{\gamma\eta IT} + 126\gamma^2 L^2 I^2\tilde{\nu}^2 + 144\gamma^2 L^2 I^2 P^2\tilde{\beta}^2 + 6\tilde{\delta}^2(P)$$

$$+ \left(21\gamma^2 L^2 I + 18\gamma^2 L^2 IP\rho^2 + 2\gamma\eta L\rho^2\right)\sigma^2$$

$$= \mathcal{O}\left(\frac{\mathcal{F}}{\gamma\eta IT} + \gamma^2 L^2 I^2\tilde{\nu}^2 + \gamma^2 L^2 I^2 P^2\tilde{\beta}^2 + \tilde{\delta}^2(P) + \left(\gamma^2 L^2 I + \gamma^2 L^2 IP\rho^2 + \gamma\eta L\rho^2\right)\sigma^2\right)$$

where the first term in $(a)$ is from telescoping of (C.16) and $\frac{1}{\lfloor T/P\rfloor P} \leq \frac{1}{(T/P-1)P} = \frac{1}{T-P} \leq \frac{1}{T-T/2} = \frac{2}{T}$ since $P \leq \frac{T}{2}$, and the other terms in $(a)$ are constants independent of $t_0$. $\qquad\square$

### C.4  Proof of Corollaries 3.2 and 3.3

For Corollary 3.2, we note that $\gamma\eta = \min\left\{\frac{\sqrt{\mathcal{F}}}{\sigma\rho\sqrt{LIT}}; \frac{1}{LIP}\right\}$ by the choice of $\gamma$ and $\eta$. Because $\gamma \leq \frac{1}{12LIP}$ (since $T \geq 1$) and $\gamma\eta \leq \frac{1}{LIP}$, the conditions of Theorem 3.1 hold. Then, using $\gamma\eta \leq \frac{\sqrt{\mathcal{F}}}{\sigma\rho\sqrt{LIT}}$, $\frac{1}{\gamma\eta} = \max\left\{\frac{\sigma\rho\sqrt{LIT}}{\sqrt{\mathcal{F}}}; LIP\right\} \leq \frac{\sigma\rho\sqrt{LIT}}{\sqrt{\mathcal{F}}} + LIP$, and $\rho^2 \leq 1$, we obtain the result.

For Corollary 3.3, we have $\gamma\eta = \frac{\sqrt{\mathcal{F}}}{\rho\sqrt{LIT}}$ according to the choice of $\gamma$ and $\eta$. Because $\frac{\sqrt{\mathcal{F}}}{\rho\sqrt{LIT}} \leq \frac{1}{LIP}$, we have $\gamma\eta \leq \frac{1}{LIP}$ and the conditions of Theorem 3.1 hold. The result is obtained by plugging the values of $\gamma$ and $\eta$ into the result in Theorem 3.1, and noting that $\rho^2 \leq 1$ in the last term. $\qquad\square$

## D  Additional Details and Results of Experiments

### D.1  Additional Details of the Setup

**Code.** Please visit `https://shiqiang.wang/code/fl-arbitrary-participation`

**Datasets.** The FashionMNIST dataset[3] has MIT license and a citation requirement [34]. The CIFAR-10 dataset[4] only has a citation requirement [19]. We have cited both papers in our main paper. Our

---

[3] `https://github.com/zalandoresearch/fashion-mnist`
[4] `https://www.cs.toronto.edu/~kriz/cifar.html`

experiments used the original split of training and test data in each dataset, where the training data is further split into workers in a non-IID manner according to the labels of data samples with a similarity of $5\%$, as described in the main paper.

**Motivation of the Setup.** The non-IID splitting of data to clients and the periodic connectivity are motivated by application scenarios of FL. An example of a real-world scenario is where different geographically-dispersed user groups can participate in training at different times of the day, e.g., when their phones are charging during the night. Another example is where multiple organizations have their local servers (also known as edge servers), which can participate in training only when their servers are idle, such as during the night when there are few user requests, and these servers are located in different geographical regions. Our notion of "client" is a general term that can stand for the phone or the local server in these specific scenarios.

**Availability of Clients.** In the case of *periodically available* clients, as described in the main paper, subsets of clients with different majority class labels take turns to become available. For example, only those clients with the first two majority classes of data are available in the first 100 rounds; then, in the next 100 rounds, only those clients with the next two majority classes of data are available, and so on. The majority class of each client is determined by the non-IID data partitioning. There are 10 classes in total, so a full cycle of participation of all clients is 500 rounds. To avoid inherent synchronicity across different simulation instances, we apply a random offset in the participation cycle at the beginning of the FL process. With this random offset, the number of rounds where the first subset of clients participate can be decreased to less than 100. For example, the first subset of clients may participate for 37 rounds (an arbitrary number that is less than or equal to 100), the second subset of clients then participates for the next 100 rounds, and so on. This random offset varies across different simulation instances. It represents the practical scenario where the FL process can start at any time in the day. In each round, $S = 10$ clients are selected to participate, according to a random permutation among the currently available clients. The selection of a small number of clients among the available clients can be due to resource efficiency considerations, for instance.

In addition to periodic availability, we also consider a setting where all the $N = 250$ clients are *always available*, and a subset of $S = 10$ clients are selected to participate in each round. Here, we consider two participation patterns. For independent participation, the subset of clients are selected randomly according to a uniform distribution, independently across rounds. For regularized participation, the clients are first randomly permuted and then $S$ clients are selected sequentially from the permuted sequence of clients, so that all the $N$ clients participate in FL once in every $\frac{N}{S}$ rounds.

**Hyperparameter Choices.** The initial local learning rates of $\gamma = 0.1$ and $\gamma = 0.05$ for FashionMNIST and CIFAR-10, respectively, were obtained from a learning rate search where the model was trained using $\gamma \in \{0.5, 0.1, 0.05, 0.01, 0.005, 0.001, 0.0005, 0.0001\}$ while fixing $\eta = 1$. We considered the case of always available clients for this learning rate search. We chose the learning rate that gave the smallest training loss after $2,000$ rounds of training for FashionMNIST and $4,000$ rounds of training for CIFAR-10, where we observed that the best learning rate is the same for both independent and regularized participation with each dataset. The reason of considering always available clients for tuning the initial learning rates is because this is the ideal case which allows the model to make big steps in training, so it is reasonable to use these learning rates as starting points even in the case of periodic availability. The number of rounds that use the initial learning rate in the case of periodic availability was determined so that the model's performance no longer improves substantially afterwards, if keeping the same initial learning rate.

The subsequent learning rates for the methods shown in Figure 1, with periodically available clients, are found from a search on a grid that is $\{1, 0.1, 0.01, 0.001, 0.0001\}$ times the initial learning rate, as described in the main paper. The optimal learning rates found from this search are shown in Table D.1.

We fix the number of local updates to $I = 5$ and use a minibatch size of 16 for local SGD at each participating client. These parameters were not tuned specifically, since the relative difference between different methods will likely follow the same trend for different $I$ and minibatch sizes. Because $I$ is a controllable parameter in our experiments, we consider a minor extension (compared to our discussion in Section 5) in the "wait-*" methods in our experiments, where each client performs $I$ local iterations (instead of one iteration in the discussion in Section 5) before starting to wait. The intuition remains the same as what has been discussed in Section 5. The reason for this extension is because the "wait-*" methods defined in this way is equivalent to the case where all the clients are

Table D.1: Optimal learning rates found from grid search

| Method | FashionMNIST | CIFAR-10 |
|---|---|---|
| wait-minibatch | $\gamma = 0.1$ | $\gamma = 0.05$ |
| wait-full | $\gamma = 0.1$ | $\gamma = 0.05$ |
| Algorithm 1 without amplification ($\eta = 1$) | $\gamma = 1 \times 10^{-5}$ | $\gamma = 5 \times 10^{-5}$ |
| Algorithm 1 with amplification ($\eta = 10, P = 500$) | $\gamma = 1 \times 10^{-5}$ | $\gamma = 5 \times 10^{-6}$ |

available but the total number of rounds is reduced by $P$, so it has an easily interpretable meaning in practice. In addition, changing $I$ can change the performance of all methods (including Algorithm 1, both with and without amplification, and the "wait-*" methods), while their relative difference can remain similar. We do not study the effect of $I$ and minibatch size in details in this work.

**Model Architecture.** The CNN architectures used in our experiments are shown in Table D.2, where we use Kaiming initialization [15] for all the ReLU layers. All convolutional and dense layers, except for the last layer, use ReLU activation.

Table D.2: CNN architectures

| FashionMNIST | CIFAR-10 |
|---|---|
| Convolutional (input: 1, kernel: $5 \times 5$, padding: 2, output: 32) | Convolutional (input: 3, kernel: $5 \times 5$, padding: 2, output: 32) |
| MaxPool (kernel: $2 \times 2$, stride: 2) | MaxPool (kernel: $2 \times 2$, stride: 2) |
| Convolutional (input: 32, kernel: $5 \times 5$, padding: 2, output: 32) | Convolutional (input: 32, kernel: $5 \times 5$, padding: 2, output: 64) |
| MaxPool (kernel: $2 \times 2$, stride: 2) | MaxPool (kernel: $2 \times 2$, stride: 2) |
| Dense (input: $1,568$, output: 128) | Dense (input: $4,096$, output: 512) |
| Dense (input: 128, output: 10) | Dense (input: 512, output: 128) |
| Softmax | Dense (input: 128, output: 10) |
| | Softmax |

**Collecting Results.** The plots in Figures 1–2 and all the figures showing additional results in the next section were obtained from simulations with 10 different random seeds for FashionMNIST and 5 different random seeds for CIFAR-10. For the best viewing experience, we further applied moving average over a window length of $3\%$ of the data points (on the $x$-axis) in the loss and accuracy results. In all the figures, the solid lines and the shaded areas show the average and standard deviation values, respectively, over the moving average window and all simulation runs for each configuration.

**Computational Resources.** The workload of experiments was split between a desktop machine with RTX 3070 GPU and an internal GPU cluster. On RTX 3070, the FashionMNIST experiment with $300,000$ training rounds takes about a day to complete for a single simulation instance; the CIFAR-10 experiment with $600,000$ training rounds takes about three days to complete for a single simulation instance. The RTX 3070 has memory that can support simulation with 5 different random seeds (i.e., 5 different instances) at the same time. On the internal GPU cluster, the running times are slightly shorter since it has better-performing GPUs. We also note that the "wait-*" methods run faster than Algorithm 1 in simulation because a lot of rounds are skipped, but this does not mean that they can train models faster in practice, because we consider the number of training rounds as an indication of physical time elapse. For example, in the case of periodic availability, each subset of clients is available for 100 rounds in a cycle. This means that 100 rounds of training can take place during the physical time duration that these clients are available. Algorithm 1 chooses to actually train models during this time. The "wait-*" methods choose not to train in some of the rounds, but they still need to wait until the time has passed and the next subset of clients becomes available, where the time here is measured by the number of training rounds that includes those rounds that may be skipped by the "wait-*" methods.

## D.2 Additional Results

In the following, we present some additional results obtained from experiments.

### D.2.1 Always Available Clients

We first consider the case of always available clients, where a subset of $S = 10$ clients are selected in an independent or regularized manner, as discussed in Section D.1. We compare the results of independent and regularized participation in Figure D.1. The observation is that both methods give similar performance, with regularized participation having a marginal performance improvement. As our theory predicts (see Table 1), both independent and regularized participation can guarantee convergence, where regularized participation gives a slightly better convergence rate.

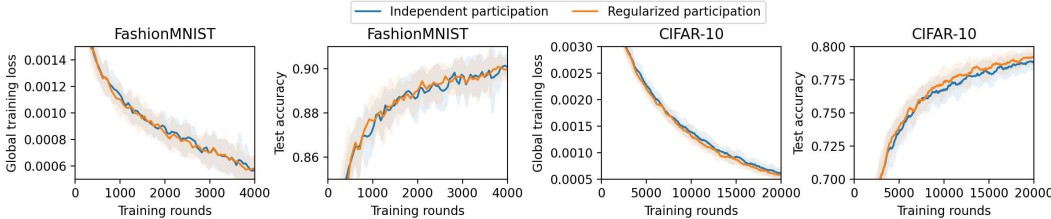

Figure D.1: Comparison between independent and regularized participation in the case of always available clients.

### D.2.2 Periodically Available Clients

**Different Learning Rates and Amplification Factors.** Next, we consider periodically available clients as in the main paper. We study the performance of Algorithm 1 with different learning rate and amplification factor configurations. The results are shown in Figures D.2 and D.3.

Figure D.2 shows the results with different combinations of the local learning rate $\gamma$ and amplification factor $\eta$. We note that $\eta = 1$ corresponds to the case without amplification and $\eta = 10$ corresponds to the case with amplification (and with an amplification factor of 10). We can observe that when $\gamma$ is large, the training is generally unstable with high fluctuation, high loss, and low accuracy. Reducing $\gamma$ improves the performance. According to our theory, $P = 500$ gives a small $\tilde{\delta}^2(P)$ with this experimental setup, because the cycle of all clients being available once is 500 rounds. On the other hand, a small $\tilde{\delta}^2(P)$ can be achieved with a much smaller $P$ in the case of always available clients (in Section D.2.1). When $P$ gets large, our theory suggests that the learning rate $\gamma$ should become small in order to guarantee convergence. This explains why non-convergence is observed in Figure D.2 for large $\gamma$, while convergence is observed for small $\gamma$. For the smallest $\gamma$ in each case, i.e., $\gamma = 10^{-5}$ for FashionMNIST and $\gamma = 5 \times 10^{-6}$ for CIFAR-10, we can see that amplification with $\eta = 10$ further improves the performance over no amplification ($\eta = 1$).

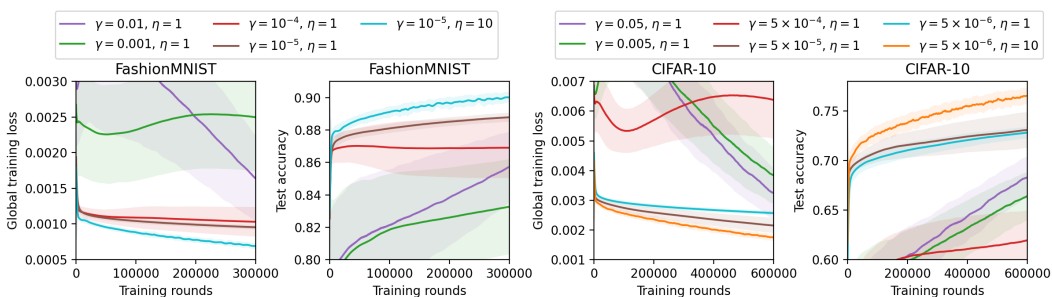

Figure D.2: Algorithm 1 with different learning rates in the case of periodically available clients. Note: FashionMNIST with $\gamma = 0.1$ is not shown in the plots due to NaN values observed when the number of training rounds is large.

In Figure D.3, we further investigate the impact of different amplification factors $\eta$, while fixing $\gamma = 10^{-5}$ for FashionMNIST and $\gamma = 5 \times 10^{-6}$ for CIFAR-10 as in the case of $\eta = 10$ in Figure D.2. We observe that the cases with $\eta = 5$, $\eta = 10$, and $\eta = 15$ have similar performance, especially when getting closer to convergence. The performance gets worse when $\eta$ is too small (e.g., $\eta = 2$), because the advantage of amplification is not fully attained in this case. When $\eta$ is too large (e.g., $\eta = 20$), we observe higher fluctuation in the loss and accuracy values, which also leads to lower performance. For both datasets and their corresponding models considered in our experiments, we observe that the cases of $\eta = 5$, $\eta = 10$, and $\eta = 15$ all give reasonably good performance. This suggests that a coarse choice of $\eta$ based on experience can be sufficient in practice, without the need of fine-grained tuning.

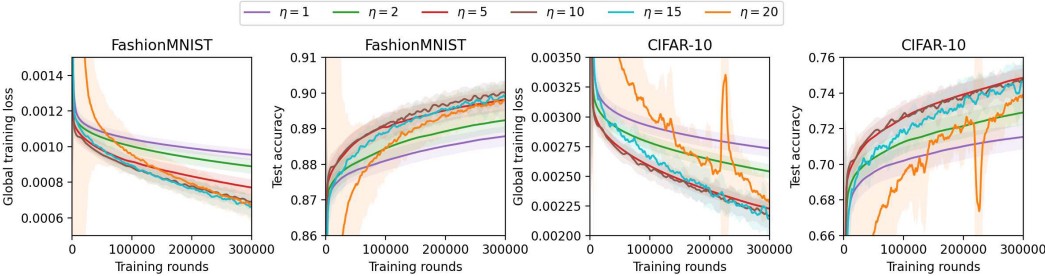

Figure D.3: Algorithm 1 with different amplification factors $\eta$, while fixing $\gamma = 10^{-5}$ for FashionMNIST and $\gamma = 5 \times 10^{-6}$ for CIFAR-10.