# OpenReview forum: "A Unified Analysis of Federated Learning with Arbitrary Client Participation"
_NeurIPS.cc/2022/Conference — NeurIPS 2022 Accept_

### Official Review · Reviewer_aHyK · 2022-06-28

**Rating:** 7
**Confidence:** 4
**Soundness:** 3 good
**Presentation:** 3 good
**Contribution:** 3 good

**Summary:**

This paper mainly provides a general convergence analysis of federated learning when there is arbitrary client participation and proposes an amplification method to update gradients in federated learning under the arbitrary participation of clients. More specifically, there are four main client participation patterns that the authors consider in their analysis: regularized, ergodic participation, mixing, and independent. The paper shows that under these participation patterns, their proposed application method for federated learning can achieve convergence to zero, and also some patterns can match the linear speed-up in the convergence rate (shown in prev. works) with the increase in the number of participating clients. The experiments on FMNIST and CIFAR-10 are presented to show validation of the speed-up that the proposed method can achieve.

**Questions:**

I have questions related to the three weaknesses mentioned in the section above. In short the following three questions:
- What is the intuition for the amplification method in algorithm 1?
- How does the hyperparameter $\eta$ effect the algorithm performance? How sensitive is the algorithm to this parameter? Is there an ablation study that can show this?
- What are some realistic scenarios that represent the four main different client participation patterns?

**Limitations:**

The authors have clearly stated the limitations of their work.

**Strengths And Weaknesses:**

*Strengths:
In general, the paper is well written and provides meaningful insight into federated learning with a unified convergence analysis for arbitrary client participation. Specifically, these are the main strength of the paper:
- The paper theoretically and empirically addresses an important problem of federated learning which is arbitrary client participation. While many work in federated learning considers a fixed client participation pattern, in real implementations of cross-device federated learning, clients can leave or join arbitrarily depending on their circumstances. There are not much work that has a unified analysis of different patterns (including stochastic) of client participation, and this paper contributes to the federated learning community largely by presenting such analysis.
- The paper compares its analysis with relevant other work including [11], [27] referenced in the paper as well as the other work which shows linear speedup in convergence with partial client participation in federated learning [31]. It is interesting to see in what scenarios for arbitrary client participation we can achieve a similar linear speedup which the paper thoroughly provides insights on.
- Although the contribution of this paper is more on the theoretical side, it proposes an interesting amplification method (line 11-14 in Algo. 1) which does not really need additional computation/communication at the client side, and only requires additional memory saving at the server side (which in general is a plus for federated learning). This method also achieves 0 convergence error in some client participation patterns which is interesting and validated in the experiments.


*Weaknesses:  In general, the paper has several weaknesses but they do not surpass the strength of the paper in my opinion. Here are some of the weaknesses:
- The intuition behind the amplification method in Algorithm 1 is unclear. Although it is a strength that using this method can assist in achieving 0 convergence error in the analysis, the insight/reason for this is unclear. The paper also lacks a bit in explaining the main intuition behind algorithm 1.
- In addition to the point above, the $\eta$ seems like a very important hyperparameter that can largely affect the performance/convergence of the algorithm. However, the clear effect of these hyperparameters has not been viewed much either theoretically or empirically throughout the paper. Perhaps having some ablation study on the $\eta$ can help further understand the implications/effectiveness of the proposed algorithm 1.
- The four main different client participation patterns that the paper considers are quite general, partly I assume is because they are defined to be applied to the theoretical analysis. It would be more interesting if the authors gave some connection with the arbitrary client participation patterns that exist in the real-world federated learning applications such as periodic shifts [6-7] (as cited in the paper).

---

> ### Author Response · Authors · 2022-08-02
> **Response to Reviewer aHyK**
>
> Thank you very much for your detailed review and questions. We answer each of your questions in the following.
>
> 1. In essence, the advantage of amplification is that it increases the importance of the aggregated updates made by clients that participate in different rounds. Consider the case where different groups of clients have different data distributions, and only one group participates in each round, such as in the periodic availability setting used in our experiments. In this case, the update direction of model parameter $\mathbf{x}$ in a single round may divert from the direction towards the global optimum. However, the accumulated updates by a more representative set of clients over $P$ rounds will be more likely towards the global optimum (or a neighborhood of it). By amplifying such accumulated updates after every $P$ rounds, we can let the solution variable $\mathbf{x}$ move much faster towards the true (global) optimum.
>
>     There is a brief mentioning about this insight at the end of Section 2. We also recognize that this intuition can be explained better with a motivating example and figures showing the trajectories of parameter updates under different settings, so we have added such a motivating example in Appendix B.1 (on page 2-3 in the appendix PDF file, included as part of the supplementary zip file), which includes Figures B.1 and B.2. Please kindly refer to Appendix B.1 for more details.
>
> 2. Theoretically, the choices of $\gamma$ and $\eta$ in our Corollaries 3.2 and 3.3 give some useful insights. For example, a smaller $\gamma$ and larger $\eta$ may be preferred when $P$ gets large, and vice versa. We agree that it would be nice to also study the effect of $\eta$ empirically. To this end, we have run a new experiment with different $\eta$ while keeping $\gamma$ unchanged. The results have been added in Figure D.3 of Appendix D.2.2 (on page 16-17 in the appendix PDF file).
>
>     From the results (see Figure D.3 for details), we observe that the cases with $\eta=5$, $\eta=10$, and $\eta=15$ have similar performance, especially when getting closer to convergence. The performance gets worse when $\eta$ is too small (e.g., $\eta=2$), because the advantage of amplification is not fully attained in this case. When $\eta$ is too large (e.g., $\eta=20$), we observe higher fluctuation in the loss and accuracy values, which also leads to lower performance. For both datasets and their corresponding models considered in our experiments, we observe that the cases of $\eta=5$, $\eta=10$, and $\eta=15$ all give reasonably good performance. This suggests that a coarse choice of $\eta$ based on experience can be sufficient in practice, without the need of fine-grained tuning.
>
> 3. The case of periodic shifts is mostly related to *regularized participation*, where the clients participate equally over the entire period, but only a subset of clients from a specific population (and with a specific data distribution) may participate in each "phase" that includes a certain number of consecutive rounds. In practice, this "equal participation" only needs to hold approximately, and our empirical results at the end of Appendix D.2.2 (Figure D.4, on page 17 in the appendix PDF file) show that our Algorithm 1 can still give good performance even if $P$ is less than the full cycle (period) of participation.
>
>     Among the stochastic participation patterns, *ergodic* is the most generic. Intuitively, it says that the participation weights of different clients are the same when averaged over a long enough time. This can represent a cross-device FL scenario, where the participation of each client at any time instance can be highly random, but the long-term average statistics of clients are the same. The case of *strongly mixing* participation includes the special case where the participation follows a Markov chain, i.e., if the client is currently unavailable, there is a certain (possibly small) probability that it will become available in the next round, and vice versa. This can also represent a cross-device FL setting, where the devices get connected and disconnected over time, and there is some randomness in the exact time it gets connected or disconnected. Finally, the case of *independent* participation may represent a cross-silo FL setting, where the reason for not participating in a certain round is not because the client is disconnected for an extended period of time, but rather because the client has some other higher-priority tasks to run so that it cannot participate in all rounds.
>
> Thank you again!

---

> > ### Comment · Reviewer_aHyK · 2022-08-08
> > **Thank you for your response!**
> >
> > All my questions were fully answered during my rebuttal. I keep my score.

---

### Official Review · Reviewer_m3NC · 2022-07-11

**Rating:** 7
**Confidence:** 4
**Soundness:** 3 good
**Presentation:** 3 good
**Contribution:** 3 good

**Summary:**

The paper considers federated learning with arbitrary patterns of client participation and provides a novel analysis. Several participation patterns have been studied and analyzed.

**Questions:**

1. In all the results, it is assumed that $P \leq T/2$. And for the $\delta^2$ term to be zero, the average participation weights need to be equal over any $P$ interval. Will these results be able to accommodate the case when $N$ is huge, such that a client will likely only participate once?
2. With $\sigma = $ in Corollary 4.3, the bound reduces to $O(1/T)$, which is what one gets with gradient descent. However, the bound in Corollary 4.7 does not reduce to $O(1/T)$ in this special case. Can the authors comment on this difference, and is there a way to address apparent sub-optimality?
3. I'm not sure if I follow the reasoning in the remark following Corollary 4.8, explaining how $I$ need not be small. Can you elaborate?

**Limitations:**

The main limitation of the paper for me is the large $N$ case, where Proposition 4.2 may be difficult to satisfy. To get a decent convergence rate, $P$ will have to be smaller than $\sqrt T$ (see Corollary 3.2, 3.3). However, for large $N$, each client may not even be sampled in each $P$ length interval.

**Strengths And Weaknesses:**

Strengths:
1. The paper is largely well-written and provides adequate intuition and explanation for the math introduced.
2. The paper makes a relevant and significant contribution, addressing the problem of arbitrary client participation.

Weaknesses:
1. Although the authors have made a good effort toward explaining all the math, in quite a few places I found myself scrolling back and forth to remind myself of the terms being discussed. I understand that the authors were working under a hard page limit. Therefore, if accepted, I would suggest the authors expand the description, especially in Sections 4.2 and 5. It might even help to remind the authors what some terms mean, even if they are defined earlier.

---

> ### Author Response · Authors · 2022-08-02
> **Response to Reviewer m3NC**
>
> Thank you very much for your helpful comments and detailed questions. We will certainly use the additional page allowed for the camera-ready version and include more descriptions in the final paper if our paper gets accepted.
>
> Our answers to your questions are as follows.
>
> 1. We have briefly discussed the practical aspects related to $P$ at the beginning of Section 5. There is also more discussion at the end of Appendix D.2.2 (on page 17 of the appendix PDF file, included as part of the supplementary zip file). We answer this question in two parts as follows.
>
> - First, in practice, the notion of a single client can be extended to a group of clients with similar data distribution. As long as there is at least one client from each group that participates within $P$ rounds, the same intuition obtained from our theory still holds. Connecting this to our theory, the case of group participation here can give a small $\delta$, because the average of $\nabla F_n(\mathbf{x})$ over all participating clients from all groups (within $P$ rounds) will be close to the global gradient $\nabla f(\mathbf{x})$, although the difference is not necessarily zero. In a practical setting with a very large user pool, there usually exist many users that have similar characteristics to each other, so their data will be also similar and this intuition holds.
>
> - Second, and perhaps more importantly, our empirical results in Appendix D.2.2 and its Figure D.4 show that we can still achieve good performance even if we set $P$ to a value that is less than the full cycle of client participation. This suggests that, in practice, the parameter $P$ in our algorithm may not need to be very large. As long as the subset of clients that participate in $P$ rounds are more representative of the overall data distribution than the (usually much smaller) subset of clients that participate in a single round, amplifying the updates every $P$ rounds can be useful.
>
> 2. We believe that you meant Corollary 4.8 instead of Corollary 4.7, since 4.7 in the paper is not a corollary. The main intuition is that, in the case of regularized participation (Corollary 4.3), the aggregated participation weights of different clients over $P$ rounds are equal. Hence, there is no randomness caused by the discrepancy in participation weights of clients, so the dominant term $O(1/\sqrt{T})$ becomes zero if we do not have randomness in gradient computation (i.e., $\sigma=0$) either. In the case of stochastic participation (Corollary 4.8), even if $\sigma=0$, there is still randomness due to the random participation of clients. This randomness in client participation has a similar effect as the noise in stochastic gradient computation, in the sense that we may consider each client as a minibatch in SGD, which holds even if the data is IID across clients. For this reason, the term $O(1/\sqrt{T})$ remains even if $\sigma=0$ in stochastic participation.
>
> 3. There is some additional discussion on this matter in Appendix B.3 (on page 3-4 in the appendix PDF file). In essence, what we mean here is that the optimal solution $\mathbf{x}^*$ to the global objective $f(\mathbf{x})$ does not change if we multiply the objective $f(\mathbf{x})$ by an arbitrary constant $a>0$. In this way, if $0<a<1$, then multiplying $f(\mathbf{x})$ by $a$ makes the coefficient $\sqrt{L\mathcal{F}}$ smaller, which in turn allows a larger maximum value for $I$ in Corollary 4.8. This scaling operation does not change the optimal solution $\mathbf{x}^*$, i.e., if $\mathbf{x}^*$ minimizes $a f(\mathbf{x})$, then it also minimizes $f(\mathbf{x})$, and vice versa. In this way, Corollary 4.8 can provide a convergence bound for the scaled objective $a f(\mathbf{x})$ for any given $I$, when choosing the constant $a$ properly. However, we acknowledge that $a f(\mathbf{x})$ and $f(\mathbf{x})$ are not exactly the same, so use the wording "potentially allow arbitrarily large $I$" in the main paper and mention this aspect as possible future work in Appendix B.3.
>
> Thank you again!

---

> > ### Comment · Reviewer_m3NC · 2022-08-08
> > **Thanks!**
> >
> > I thank the authors for addressing all my questions.

---

### Official Review · Reviewer_Th7z · 2022-07-12

**Rating:** 7
**Confidence:** 4
**Soundness:** 3 good
**Presentation:** 3 good
**Contribution:** 3 good

**Summary:**

This paper studies the partial participation aspect in local SGD method. In previous works uniform sampling of clients is usually assumed, which is not close to real situation in practice. In this work authors introduce a new general framework for analyzing and understanding different strategies of partial participation.

The framework allows to decouple terms such that it is possible to analyze effect of partial participation separately. Authors firstly provide general results and then analyze some special cases and reflect results.

In the end of the paper a wide range of experimental results was provided.

**Questions:**

Can you please explain if it is possible to do such analysis for local methods which can tackle heterogeneity? (https://arxiv.org/pdf/1910.06378.pdf, https://arxiv.org/abs/2202.09357)

Is it possible to extend the analysis for the case when bounded heterogeneity is not used?

**Limitations:**

There are two main limitations:

1) No analysis for convex objectives

2) Bounded heterogeneity assumption

3) No analysis for local methods that can tackle client drift (heterogeneity): SCAFFOLD, ProxSkip, FedLin.

Overall, contribution is quite good and despite all limitations I lean towards to accept.

**Strengths And Weaknesses:**

Strengths:

The provided framework is general and it covers many practical examples. The paper is well-written in general, all assumptions and definitions are clearly stated. All formulations of theorems are clear.

Theoretical analysis seems to be correct. I checked the appendix and I did not find mistakes, but I might miss something.

Obtained results are interesting and they lead to new prospectives and insights.

The experimental comparison is meaningful and illustrative.

Weaknesses:

The analysis relies on Bounded gradient divergence assumption. It means that level heterogeneity is limited. This assumption contradicts to experimental design, where they "partition the data into clients so that each client has data of one majority class label." It would be interesting to see analysis without bounded heterogeneity.

The analysis is done only for non-convex objectives. It would be interesting to see analysis for strongly convex and general convex objectives. Also it is interesting to study PL condition.

---

> ### Author Response · Authors · 2022-08-02
> **Response to Reviewer Th7z**
>
> Thank you very much for your insightful comments and questions. Our response to your questions and comments are as follows.
>
> **[Bounded Heterogeneity]** Regarding bounded gradient divergence, we first would like to explain that this does not necessarily imply that the bound needs to be very small. Our theory holds as long as the gradient divergence bound is finite, which is indeed the case in our experiments although each client has data of only one majority class label. Nevertheless, we understand that the main point of your question seems to be whether we can use a weaker assumption in our analysis. To this end, we would like to point out a few aspects as follows.
>
> - As a first step towards understanding the effect of arbitrary client participation, our focus in this paper is on a generalized variant of FedAvg or local SGD algorithms *without* the use of control variates. With partial client participation in this type of algorithms, it is natural that the convergence is related to the degree of heterogeneity. The reason is that, even if clients participate according to uniform sampling, there is additional noise in the gradient updates caused by the fact that not all clients participate in all rounds, and the degree of such noise is related to the difference in the gradients of local objectives. For this reason, the analysis on FedAvg and local SGD in other works also have some form of assumption on bounded heterogeneity.
>
> - Because we consider arbitrary participation patterns in our analysis, a key challenge in our work is that, in each round, the clients' contributions to parameter updates can be biased. This is in stark contrast to existing works that consider uniform and independent client sampling across rounds, in which case the clients' contributions are unbiased. This biasedness causes additional difficulties to our analysis, and our assumption on bounded gradient divergence facilitates such analysis.
>
> - Our gradient divergence bound provides meaningful insights, by decomposing it into multiple parts as in Assumption 3'. This gives explainability on how different aspects of heterogeneity affect the convergence.
>
>
> **[Convex/PL Objectives]** We focus on general non-convex objective functions in this paper, and have obtained new insights on how different participation patterns and parameter settings (such as $P$ and $\eta$) affect the convergence. An extension to convex or PL objectives can be considered in future work, where our overall methodology should be still applicable, but it needs to be tailored to align with the specific convexity assumptions.
>
>
> **[Local Methods]** We believe that our current results for (generalized) FedAvg can inspire a range of future works that analyze more sophisticated algorithms. Our core ideas, such as amplification over multiple rounds and capturing the effect of participation by a single term in the convergence bound, can potentially play an important role in the analysis of more advanced algorithms too. In the case of local methods where the heterogeneity is compensated by the control variates, the convergence bound no longer depends on the heterogeneity. However, the lag of control variate updates due to partial client participation has an impact on convergence, and existing works such as SCAFFOLD have only analyzed the case of uniform and independent client sampling in each round. To analyze arbitrary participation in such local methods, a similar decoupling of participation-related parameters from the rest of the convergence bound can be very useful, where ideas in our analysis may be leveraged. While it is not easy to speculate the exact steps of such an analysis, this is certainly a very interesting topic worth studying in the future.
>
> Thank you again!

---

> > ### Comment · Reviewer_Th7z · 2022-08-06
> > **Thank you for your response!**
> >
> > >Regarding bounded gradient divergence, we first would like to explain that this does not necessarily imply that the bound needs to be very small. Our theory holds as long as the gradient divergence bound is finite, which is indeed the case in our experiments although each client has data of only one majority class label. Nevertheless, we understand that the main point of your question seems to be whether we can use a weaker assumption in our analysis. To this end, we would like to point out a few aspects as follows.
> >
> > In (https://arxiv.org/pdf/1909.04746.pdf) analysis is done without bounded heterogeneity assumptions. Yes, it depends on level of heterogeneity:
> > $$\sigma_{diif}^2 = \frac{1}{M} \sum_{m=1}^{M} \mathbb{E} \left[\left\|\nabla f_m\left(x_*, z_m\right)\right\|^2\right] $$. I was wondering if it is possible to make similar analysis without bounded heterogeneity assumption.
> >
> > Overall, I agree that my comments are mostly ideas for future work, so I believe that this paper deserves to be accepted and I keep my initial score (7).

---

> > > ### Author Response · Authors · 2022-08-08
> > > **Additional Response to Reviewer Th7z  (Some More Thoughts on Heterogeneity)**
> > >
> > > Many thanks for your reply and positive opinion! We would like to share a bit more thoughts on heterogeneity as follows.
> > >
> > > 1. The paper that you mentioned in your last comment studies convex and $L$-smooth objectives. Such objective functions have some useful properties related to Bregman divergence, which are key to making the analysis based on heterogeneity at the global minimum point $\mathbf{x}^\ast$ possible. For such convex and smooth objectives, there is actually a relation between gradient divergence at any point $\mathbf{x}$ and the norm of gradient at $\mathbf{x}^\ast$, as shown in Section 2.2 of [a]. However, these properties do not hold for non-convex objective functions, and our paper considers the non-convex case.
> > >
> > > 2. Another way to look at this is that, when the objective functions are convex and smooth, it is possible to derive a recursive update relation that shows $\Vert \mathbf{x}_{t+1} - \mathbf{x}^\ast \Vert^2$ is upper bounded by a quantity related to $\Vert \mathbf{x}_t - \mathbf{x}^\ast \Vert^2$ and some other terms. This relation includes $\mathbf{x}^\ast$ as a "reference point". Informally, this makes it possible to include other terms related to $\mathbf{x}^\ast$, which can be combined with the $\Vert \mathbf{x}_t - \mathbf{x}^\ast \Vert^2$ term after some manipulation. In contrast, the recursive update in the analysis of non-convex objectives is directly derived from smoothness, because we do not have additional convexity assumptions. These update relations *do not* include the common reference point $\mathbf{x}^\ast$. Therefore (again, informally), in the non-convex case, it is difficult include a term related to $\mathbf{x}^\ast$ (or any other fixed value of $\mathbf{x}$) on one side of the inequality with the hope that it diminishes after the recursion is unrolled.
> > >
> > > 3. Because of the possibly biased updates caused by arbitrary participation in our setting, a simple average of heterogeneity over all the clients, as in both [a] and [b] (the paper that you mentioned), is not directly applicable to our analysis. As seen in Section 2.2 of [a] and the proof of Lemma 11 in [b], this averaging step is necessary to connect the Bregman divergence of local objectives to that of the global objective. This suggests that, even in the case of convex and smooth objectives, it may be difficult to use heterogeneity at $\mathbf{x}^\ast$ in the analysis of our algorithm with arbitrary participation.
> > >
> > > Due to these reasons, we believe it is extremely difficult (if not impossible) to analyze the convergence of our algorithm (with arbitrary participation) based on the heterogeneity at a fixed point $\mathbf{x}^\ast$, especially when considering non-convex objectives as in our paper. If our algorithm is extended to include variance reduction across clients, then an analysis with relaxed heterogeneity assumption could be possible, as mentioned in our previous comment. However, the result of such an extension will be a different algorithm and it can be a direction for future work.
> > >
> > > Thank you again!
> > >
> > > \
> > > [a] Stich and Karimireddy. The Error-Feedback Framework: Better Rates for SGD with Delayed Gradients and Compressed Updates. JMLR, 2020.
> > >
> > > [b] Khaled, Mishchenko, and Richtarik. Tighter Theory for Local SGD on Identical and Heterogeneous Data. AISTATS 2020.

---

### Author Response · Authors · 2022-08-09
**Brief Summary, and Thank You!**

We sincerely thank all of you for your kind review. It is great to see a unanimously positive opinion about our work. We are also very happy that you have all replied to our responses.

Since the reviewer-author discussion period is ending soon, we would like to provide a short summary and reflection on the main discussion points, based on our understanding.

- [Reviewer Th7z](https://openreview.net/forum?id=qSs7C7c4G8D&noteId=0wugga5zRYK) has thanked for our response and agrees that many of the comments are ideas for future work. The reviewer is particularly interested in whether our analysis can be done with a weaker assumption on heterogeneity. To this end, we have taken a close look at the various analytical techniques and tools. Our conclusion is summarized in our [latest comment](https://openreview.net/forum?id=qSs7C7c4G8D&noteId=V4bzLocnZgL), where we find that our current set of assumptions is likely to be minimal to make the analysis possible. The challenge in our analysis originates from the fact that we focus on arbitrary (possibly biased) client participation and non-convex objectives. In addition, we would like to point out that our work considers a generalized version of FedAvg *without* additional computation, communication, or storage at the client side and only minimal additional overhead at the server side. This by itself has practical value and advantage (which is also recognized by [Reviewer aHyK](https://openreview.net/forum?id=qSs7C7c4G8D&noteId=is9lqsc9Hx4)), while noting that our methodology may also inspire future work on more sophisticated algorithms, such as those with variance reduction.

- [Reviewer m3NC](https://openreview.net/forum?id=qSs7C7c4G8D&noteId=n8nTa8hYMoO) has confirmed that we have addressed all the questions. The reviewer brought up a very interesting point on what happens if the total number of clients $N$ is very large. We have answered this question from both theoretical and practical perspectives. In essence, even if only a small subset of clients participate in every $P$ rounds, as long as this subset is representative of the overall population, our generalized FedAvg algorithm can converge to a near-optimal point. We have also verified this insight in experiments (in Appendix D.2.2). In fact, this suggests that our overall framework and algorithm are quite generic, and they can be a useful tool for understanding and optimizing the performance of practical federated learning systems with a huge number of clients. Although this is not the primary focus of this work, it is nice to see that such possible future developments can be made based on our work. Besides this, we have also answered the other insightful questions by the reviewer.

- [Reviewer aHyK](https://openreview.net/forum?id=qSs7C7c4G8D&noteId=is9lqsc9Hx4) has confirmed that we have fully answered all the questions. The reviewer was interested in seeing the intuitions behind amplification and different participation patterns, and also the effect of the amplification factor $\eta$. We have added a motivational example in Appendix B.1 with trajectories of parameter updates both with and without amplification, new experimental results in Appendix D.2.2 for different $\eta$, and also provided more explanation in our response. We believe that, with these additions, our paper has been further improved so that it provides in-depth discussion and insights in various forms, including algorithm design, theory, experiments, and the intuition behind them.

For the final version, we will revise the paper and make sure that all the discussions are included.

In general, we believe that our work is a significant milestone towards understanding the effect of arbitrary client participation in federated learning. This problem is important from both theoretical and practical perspectives. By establishing the foundation, our results in this paper can be very impactful, and they also have the potential to inspire a range of future works on this topic.

Again, we greatly appreciate the reviewers and area chairs for your dedication and efforts put in reviewing and discussing this paper. Thank you very much!

---

### Meta-Review · Area_Chair_uTmU · 2022-08-30

**Recommendation:** Accept
**Confidence:** Certain

**Metareview:**

The paper was appreciated by all three reviewers, and all gave strong endorsement. Some of the positive comments include:

- The paper is well-written in general, all assumptions and definitions are clearly stated.
- All formulations of theorems are clear.
- Theoretical analysis seems to be correct. I checked the appendix and I did not find mistakes, but I might miss something.
- Obtained results are interesting and they lead to new prospectives and insights.
- The experimental comparison is meaningful and illustrative.
- The paper is largely well-written and provides adequate intuition and explanation for the math introduced.
- The paper makes a relevant and significant contribution, addressing the problem of arbitrary client participation.
- In general, the paper is well written and provides meaningful insight into federated learning with a unified convergence analysis for arbitrary client participation.
- The paper theoretically and empirically addresses an important problem of federated learning which is arbitrary client participation. While many work in federated learning considers a fixed client participation pattern, in real implementations of cross-device federated learning, clients can leave or join arbitrarily depending on their circumstances. There are not much work that has a unified analysis of different patterns (including stochastic) of client participation, and this paper contributes to the federated learning community largely by presenting such analysis.
- The paper compares its analysis with relevant other work including [11], [27] referenced in the paper as well as the other work which shows linear speedup in convergence with partial client participation in federated learning [31]. It is interesting to see in what scenarios for arbitrary client participation we can achieve a similar linear speedup which the paper thoroughly provides insights on.
- Although the contribution of this paper is more on the theoretical side, it proposes an interesting amplification method (line 11-14 in Algorithm 1) which does not really need additional computation/communication at the client side, and only requires additional memory saving at the server side (which in general is a plus for federated learning). This method also achieves 0 convergence error in some client participation patterns which is interesting and validated in the experiments.

I have read the reviews, rebuttal, and also skimmed through the paper. Virtually all criticism was successfully addressed; and I would ask the authors to make sure all changes that were promised would be implemented.

I agree with the reviewers that this paper clearly passes the acceptance bar. Congratulations on such a nice paper!

AC

**Award:**

No

---

### Decision · Program_Chairs · 2022-09-14

Accept